# Size-dependent activity and selectivity of carbon dioxide photocatalytic reduction over platinum nanoparticles

Chunyang Dong[1], Cheng Lian[1], Songchang Hu[1], Zesheng Deng[1], Jianqiu Gong[1], Mingde Li[2], Honglai Liu[1], Mingyang Xing [1] & Jinlong Zhang [1]

Platinum nanoparticles (Pt NPs) are one of the most efficient cocatalysts in photocatalysis, and their size determines the activity and the selectivity of the catalytic reaction. Nevertheless, an in-depth understanding of the platinum's size effect in the carbon dioxide photocatalytic reduction is still lacking. Through analyses of the geometric features and electronic properties with variable-sized Pt NPs, here we show a prominent size effect of Pt NPs in both the activity and selectivity of carbon dioxide photocatalytic reduction. Decreasing the size of Pt NPs promotes the charge transfer efficiency, and thus enhances both the carbon dioxide photocatalytic reduction and hydrogen evolution reaction (HER) activity, but leads to higher selectivity towards hydrogen over methane. Combining experimental results and theoretical calculations, in Pt NPs, the terrace sites are revealed as the active sites for methane generation; meanwhile, the low-coordinated sites are more favorable in the competing HER.

---

[1] Key Laboratory for Advanced Materials and Institute of Fine Chemicals, School of Chemistry & Molecular Engineering, East China University of Science and Technology, 130 Meilong Road, Shanghai 200237, P. R. China. [2] Department of Chemistry and Key Laboratory for Preparation and Application of Ordered Structural Materials of Guangdong Province, Shantou University, Shantou 515063, P. R. China. These authors contributed equally: Chunyang Dong, Cheng Lian. Correspondence and requests for materials should be addressed to M.X. (email: mingyangxing@ecust.edu.cn)

The excessive emission of $CO_2$, one of the most common green-house gases, has caused global warming, which in turn gives rise to severe environmental problems. Thus, it is clearly urgent to find possible solutions to address this problem[1]. On the basis of semiconductor photocatalysis technology, $CO_2$ and $H_2O$ can be converted into some useful chemicals, such as $CH_4$, under solar light irradiation, which appears to be an ideal approach to address the above-mentioned problematic $CO_2$ emissions. However, since $CO_2$ is a stable molecule, its photocatalytic reduction (CO2PR) with $H_2O$ yielding $CH_4$ is an endothermic reaction and requires the participation of multiple electrons and protons. Hence, without modification, bare semiconductors often show poor CO2PR activity[2, 3]. Furthermore, the hydrogen evolution reaction (HER) often competes with the CO2PR for the photo-generated electrons and leads to lower CO2PR selectivity[2, 4–6]. According to previous studies, after loaded with proper cocatalysts, the performance of the photocatalyst composite can be significantly enhanced in CO2PR[7, 8]. As one of the mostly widely used cocatalyst, platinum nanoparticles (Pt NPs) are always considered as the electron trapping agents and active sites in photocatalysis[9–11]. Wang et al.[12] adopted the tilted-target sputter method for the ultrafine Pt cluster preparation (0.5–2.0 nm) and found that 1.0-nm Pt loaded composite showed the highest $CH_4$ yield due to optimal electron trapping ability; Xie et al.[13] also studied the size effect of Pt NPs with different preparation methods and found that smaller Pt NPs showed better performance in CO2PR. Our previous preliminary work has found that the particle size of Pt affects both the activity and selectivity in CO2PR[8]. Although much effort has been devoted to explaining the roles and properties of Pt in photocatalysis, there is still a lack of objective and systematic studies on the size effect of Pt NPs with a wide size range and precise size distribution in CO2PR, especially for the deep understanding of the active sites on the Pt NPs surface.

Generally, the electronic or geometric structures are altered when the particle size of Pt NPs is tuned. With the decrease in particle size, the proportion of the low-coordinated surface sites (corner or edge) increased[14, 15]. In addition, the metal oxidation states also exhibited size-dependent properties[16, 17]. Therefore, a good understanding of the size-dependent activity of Pt has been demonstrated in many catalytic reactions, such as ammonia borane dehydrogenation[14, 18], regioselective hydrogenation of quinolone[16], and the oxygen reduction reaction[15], among others. Recently, size-dependent activity and selectivity in the $CO_2$ electrocatalytic reduction reaction (CO2RR) have been reported for metal NPs, such as Pd, Au and Cu, due to the intrinsic free energy of key intermediate evolutions on different surface sites[19–21]. Inspired by these works, we attempt to explore the size effect of Pt NPs in CO2PR by studying the diversity of electronic features and geometric properties of Pt NPs.

Traditional size control methods of Pt NPs often involve variation of the amount of the Pt precursor or post deposition of presynthesized colloidal Pt NPs with controllable size on the support[12, 14, 22, 23]. However, neither of these approaches is ideal for CO2PR. For photocatalysis, loading with different amounts of Pt would cause the imbalance of light absorption of the supporting semiconductor[24, 25]; meanwhile, it also cannot tune the size of Pt NPs precisely. Smaller Pt NPs may still exist with excessive Pt loadings, and the size effect may not be reflected accurately. On the other hand, for colloidal Pt, polyvinyl pyrrolidone (PVP) was often introduced as the stabilizing agent in order to prevent the Pt NPs from aggregation; it is then difficult to remove PVP completely, and this may affect the performance of the Pt NPs[26, 27]. Therefore, two aspects of the objective investigation of the Pt size effect are challenging: precise control of the size of Pt NPs under the assumption of a constant loading

amount, and exclusion of the negative influence of the carbon impurities induced by the use of PVP during the preparation.

In this work, various Pt NPs supported on hierarchically ordered $TiO_2$–$SiO_2$ porous materials (HTSO) are synthesized via a facile acid–base-mediated alcohol reduction (ABAR) method. With the increase in the Pt NPs size, different samples are denoted as $x$PHTSO (where $x$ represents the Pt particle sizes of 1.8, 3.4, 4.3, and 7.0 nm). The variation of the geometric features and electronic properties with the size variation is carefully characterized. The size effect of Pt NPs is found to exist for both the activity and selectivity of CO2PR. Smaller Pt NPs favor charge transfer and show superior performance in both CO2PR and HER; a higher $CH_4$ selectivity is achieved by the larger Pt NPs due to their higher proportion of surface terrace sites. The changes in the active sites of Pt NPs for the CO2PR and the competing HER with the size variation are also discussed in detail.

## Results

**Synthesis and morphology of variable-sized $x$PHTSO.** Considering the consistent reaction conditions in the preparation of $x$PHTSO ($x$ = 1.8, 3.4, 4.3, and 7.0), we propose that the nucleation amount of the Pt precursor ($H_2PtCl_6·6H_2O$) is closely related to the final particle size of Pt NPs, which is adjusted by the acid or base additives in the solvent reduction process, as shown in Fig. 1a. First, the dissociated $PtCl_6^{2-}$ anions were absorbed on the surface of HTSO due to the electrostatic force, and this was then followed by the ligand-exchange process with $H_2O$ or $OH^-$ within acidic or basic media, respectively (Eqs. 1 and 2), generating the Pt complex species[28, 29]; finally, the Pt complex was reduced to Pt nuclei in the heated ethylene glycol (EG) or methanol (MeOH) solvent and grows into larger Pt NPs according to the Ostwald ripening theory[30]. In this case, the nucleation amount determines the final size of Pt NPs; therefore, in the basic environment, the $PtCl_{6-x}(OH)_x^{2-}$ can be more easily reduced, and the nucleation is faster and yields more Pt nuclei. On the contrary, the acid environment restrains the dissociation of $H_2PtCl_6$ (Eq. 3), and therefore, the nucleation of Pt is slow and fewer Pt nuclei will be formed. Consequently, with the increase in the acid concentration, larger Pt NPs will be obtained. To exclude the possibility that the size of Pt NPs is affected by the concentration of $Cl^-$ instead of the $H^+$ in the acidic environment, we simply replace the hydrochloride acid (HCl)–EG solution with the NaCl–EG solution with the same $Cl^-$ concentration in the 7.0PHTSO synthesis process. The obtained Pt/HTSO shows an average Pt NPs size of 2.0 nm rather than 7.0 nm, clarifying that the concentration of $Cl^-$ is less important for the size manipulation in our method (Supplementary Fig. 1).

$$PtCl_6^{2-} + xOH^- \rightarrow PtCl_{6-x}(OH)_x^{2-} + xCl^-. \tag{1}$$

$$PtCl_6^{2-} + yH_2O \rightarrow PtCl_{6-y}(H_2O)_y^{(2-y)-} + yCl^-. \tag{2}$$

$$H_2PtCl_6 \leftrightarrows 2H^+ + PtCl_6^{2-}. \tag{3}$$

Since the real loading amount of Pt in $x$PHTSO ($x$ = 1.8, 3.4, 4.3, and 7.0) is approximately 1.6 wt% as measured by ICP-AES (inductively coupled plasma-atomic emission spectrometry) (Table 1), decreasing the size of the Pt NPs will increase the surface density in their composite counterparts. Therefore, to study the size effect of the Pt NPs properly, the Pt NPs should be sufficiently well-dispersed on the support surface. In this case, the HTSO would be an ideal candidate due to its large surface area and 3D interconnected pore channels[8, 31]. Figure 1b–m shows the transmission electron microscopy (TEM) and high-resolution

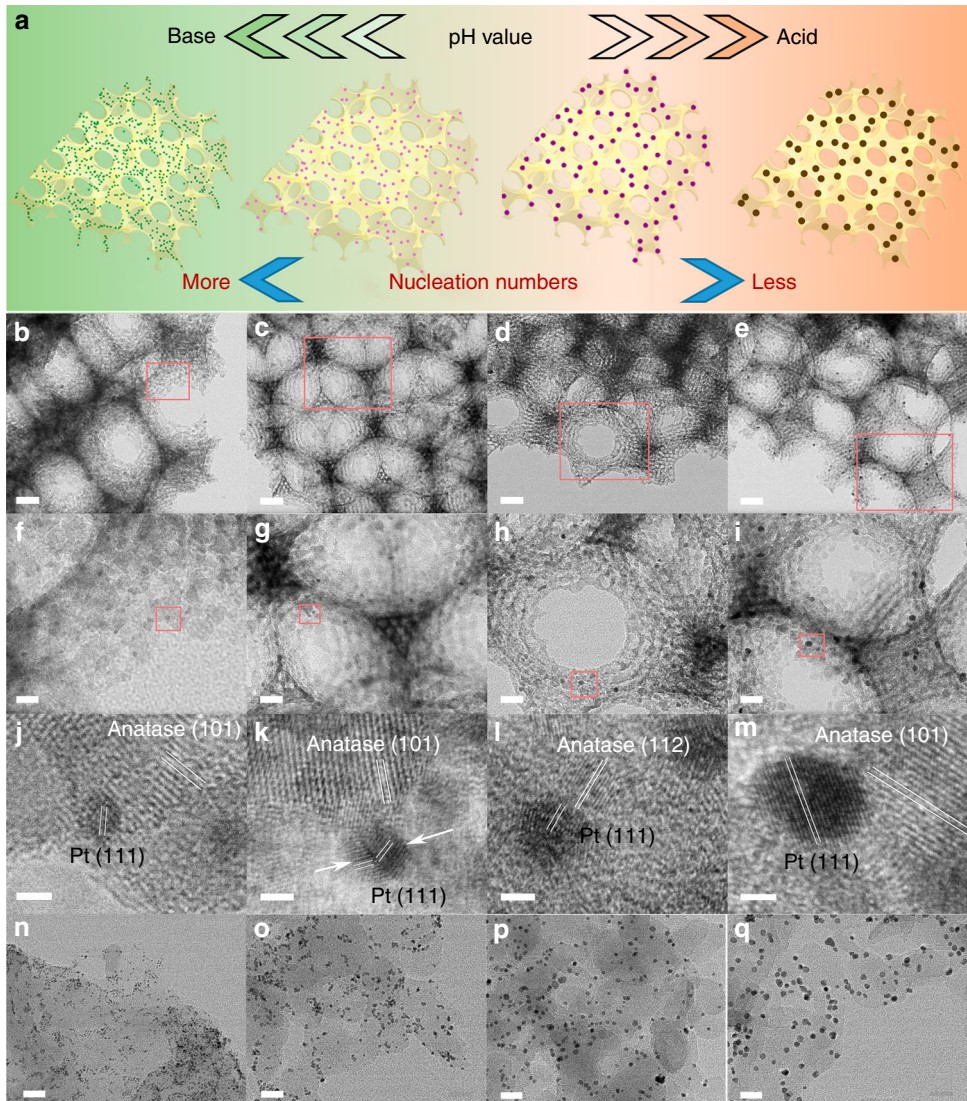

**Fig. 1** Preparation and structure characterization of the Pt nanoparticle distribution. **a** Schematic illustration of the acid–base-mediated alcohol reduction method for manipulation of the size of Pt NPs. TEM and corresponding HR-TEM images of **b**, **f**, **j** 1.8PHTSO, **c**, **g**, **k** 3.4PHTSO, **d**, **h**, **l** 4.3PHTSO, and **e**, **i**, **m** 7.0PHTSO; the red squares indicate the stepwise magnifications of the local sites. HR-TEM images of **n** 1.8 nm Pt/C₃N₄; **o** 3.4 nm Pt/C₃N₄; **p** 4.3 nm Pt/C₃N₄; and **q** 7.0 nm Pt/C₃N₄. The scale bars are 50 nm in **b**–**e**; 10 nm in **f**; 20 nm in **g**–**i**; **n**–**q**; and 2 nm in **j**–**m**

**Table 1 Characterizations of Pt NPs in the xPHTSO (x = 1.8, 3.4, 4.3, and 7.0)**

| Sample | Pt loading amount (%)[a] | Pt dispersion[b] | Pt particle size (nm) | | Pt chemical state ratio (%)[e] |
|---|---|---|---|---|---|
| | | | TEM[c] | CO pulse[d] | |
| 1.8PHTSO | 1.6 | 0.73 | 1.8 | 1.5 | $Pt^0$ (69.8), $Pt^{\delta+}$ (30.2) |
| 3.4PHTSO | 1.6 | 0.45 | 3.4 | 2.5 | $Pt^0$ (67.1), $Pt^{\delta+}$ (32.9) |
| 4.3PHTSO | 1.7 | 0.32 | 4.3 | 3.5 | $Pt^0$ (67.3), $Pt^{\delta+}$ (32.7) |
| 7.0PHTSO | 1.6 | 0.13 | 7.0 | 8.7 | $Pt^0$ (67.1), $Pt^{\delta+}$ (32.9) |

[a] Pt loading amount was determined by ICP-AES
[b] Pt dispersion: $D$(%) was calculated by using the equation: $D = $ Pt (surface)/Pt (total) $= n(CO)/n(Pt)$, where $n(CO)$ denotes the total CO adsorption amount
[c] The average size value of Pt NPs was determined by random counting of 200 Pt NPs in the TEM images
[d] Calculated using the equation: $d$ (nm) $= 1.13/D$
[e] The chemical state ratio was calculated by using the equation: $P(Pt^0) = I(Pt^0)/[I(Pt^0) + I(Pt^{\delta+})]*100\%$, where $I$ denotes the integration of the corresponding peaks

TEM images of xPHTSO (x = 1.8, 3.4, 4.3, and 7.0) with stepwise local magnifications. It is clearly seen that irrespective of the Pt NPs size, all NPs show uniform size distribution and disperse homogeneously on the entire HTSO skeleton, with no aggregation of Pt NPs found in all samples (Fig. 1b–i). The corresponding statistic obtained from the size distribution histogram is the average size value of Pt NPs in different xPHTSO samples, which we find to be 1.8, 3.4, 4.3, and 7.0 nm, respectively, with a narrow size distribution (Supplementary Fig. 2). Additionally, the interface formed by specific anatase facets and the Pt (111) facet can

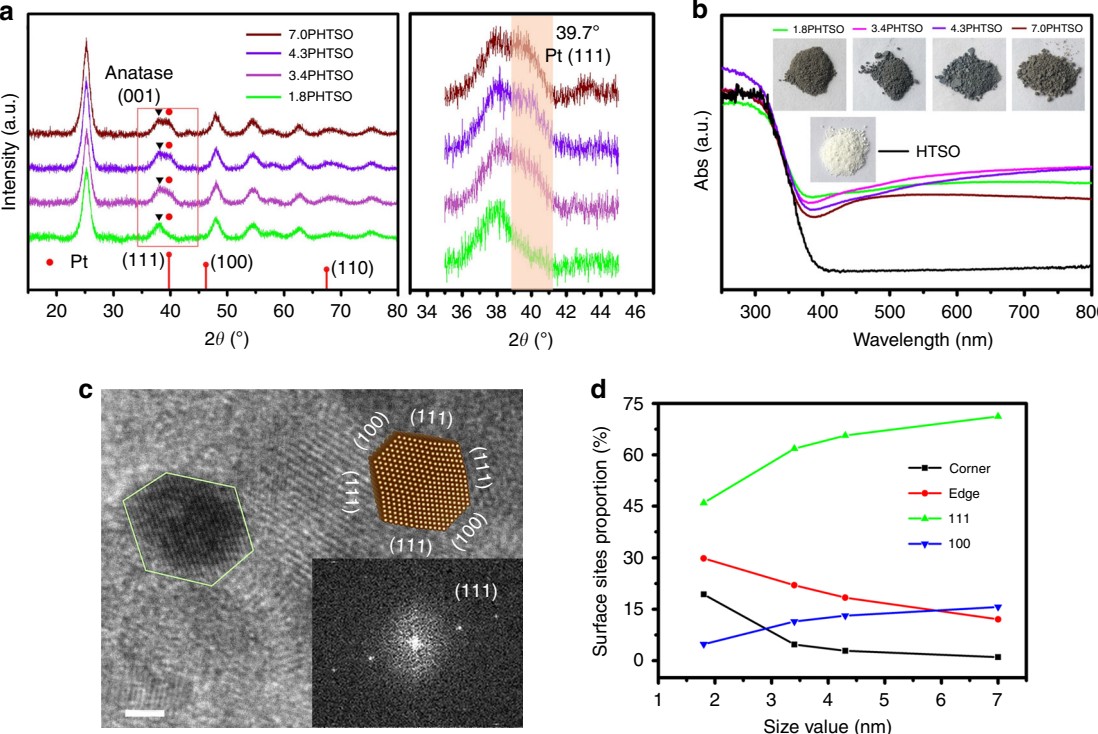

**Fig. 2** Crystallinity, surface morphology and optical properties of *x*PHTSO. **a** XRD patterns and the inset image show the local magnification from 2θ at the 35–45° range. **b** UV–Vis DRS spectra and the corresponding samples' photos (inset). **c** HR-TEM image of 7.0PHTSO, the scale bar is 2 nm; the inset image with the marked facets of Pt NPs shows the characteristic truncated-octahedron shape of this Pt particle; the inset image of the corresponding fast Fourier transform (FFT) pattern of the Pt particle. **d** Proportions of the corner, edge, Pt (111) and Pt (100) surface sites based on the standard truncated octahedron model of Pt NPs with different sizes

also be clearly observed, indicating the close contact between the Pt NPs and supporting $TiO_2$ (details in Supplementary Note 1), which enables the easy charge transfer at these interface (Fig. 1j–m). Furthermore, we adopted the ABAR strategy to achieve the highly dispersed Pt NPs on other semiconductor surfaces, such as $C_3N_4$ nanosheets (Fig. 1n–q) and commercial $TiO_2$ (P25, Supplementary Fig. 3). It can be clearly observed that the single-sized Pt NPs are highly dispersed on the surface of $C_3N_4$ and P25, and the particle size increases with the decrease in pH value. The obtained composites show the consistent size value and narrowed size distribution compared with *x*PHTSO, implying that the ABAR strategy can be easily extended to other materials.

**Geometric features and optical properties of Pt NPs**. X-ray diffractometer (XRD) was used to evaluate the overall particle sizes of Pt NPs in the *x*PHTSO composites. As the Pt NPs' size increased, as observed in the TEM images, the characteristic peak of the fcc Pt (111) facet located at $2\theta = 39.7°$ becomes sharper, indicating higher crystallinity and a larger size of Pt NPs (Fig. 2a). Additionally, the supporting HTSO also shows consistent anatase phase and peak intensity. In conjunction with the TEM analysis, this indicates that the loading of Pt NPs did not affect the crystallinity and fine structure of the HTSO. Furthermore, UV–Vis DRS analysis was adopted to characterize the light absorption features of *x*PHTSO (Fig. 2b). Compared to blank HTSO, all *x*PHTSO catalysts show enhanced visible light absorption that is mainly due to the light absorption properties of Pt. Specifically, the visual images of the 3.4PHTSO and 4.3PHTSO show dark gray color and the 1.8PHTSO and 7.0PHTSO show brown color (inset of Fig. 2b). However, the unchanged absorption edge (~400 nm) over *x*PHTSO suggests that the Pt loading cannot

change the bandgap of the supporting HTSO. Additionally, the related flat absorption spectra of *x*PHTSO in the visible light region indicate the absence of the surface plasma resonance effect over Pt NPs. Therefore, the HTSO should be the main source of photo-generated electrons and holes, while the Pt NPs act as the electron trapping agents and the active sites in CO2PR.

To objectively study the Pt NPs size effect in the CO2PR, the interference factor of surface impurities should be eliminated. To clearly observe the clean and ligand-free surface of Pt NPs, CO pulse chemisorption over *x*PHTSO was performed, as shown in Table 1. It should be noted that the CO adsorption amount is correlated with the number of surface Pt atoms. Under the same loading amount of Pt, smaller Pt NPs with a smaller bulk atom fraction should present higher CO adsorption and higher Pt dispersion ($D$) values. Additionally, in terms of catalytic reactions, the active sites over metal NPs should be uncovered and accessible to the reactant molecules. Obviously, the 1.8PTHSO with the ligand-free surface shows the highest metal dispersion value, and the calculated size of Pt NPs in *x*PHTSO ($x$ = 1.8, 3.4, and 4.3) based on the equation $d$ (nm) = 1.13/$D$ shows the same trend and similar particle size values as those obtained from the TEM results (Table 1)[27]. The sole exception is that the 7.0PHTSO shows a relatively larger size value of 8.7 nm, due to the distinct mesoporous structure of HTSO with an average pore size of around 5.5 nm (Supplementary Table 1), which confines some Pt NPs within the channels and thus blocks the CO adsorption. Generally, for the Pt NPs synthesized by the method involving PVP, the particle size calculated based on the CO chemisorption is obviously larger than that measured by the TEM or XRD characterization, due to the existence of remaining carbon impurities on the Pt surface[27]. However, in our case, the Pt particle size calculated from the chemisorption is slightly smaller

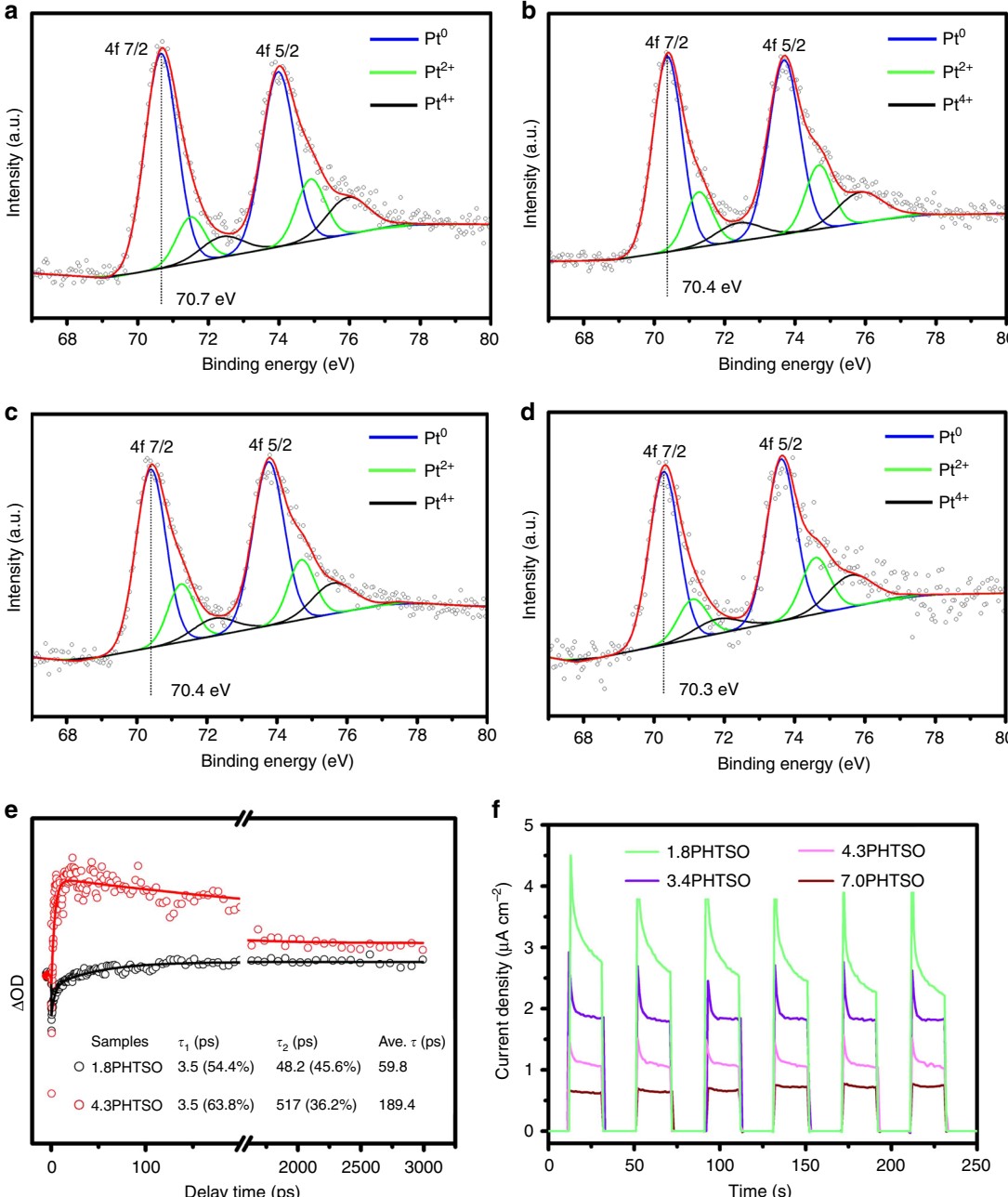

**Fig. 3** Electronic properties of *x*PHTSO. High-resolution Pt 4f XPS spectra of **a** 1.8PHTSO, **b** 3.4PHTSO, **c** 4.3PHTSO, and **d** 7.0PHTSO. **e** The kinetics of the characteristic transient absorption band observed at 350 nm in the fs-TA spectra after 310 nm excitation of samples 1.8PHTSO and 4.3PHTSO. The solid lines were the curves fitted by a two exponential. **f** Transient photocurrent response of *x*PHTSO (*x* = 1.8, 3.4, 4.3, and 7.0), where a 300 W Xe lamp is used as the light source and a 0.5 M $Na_2SO_4$ solution is used as the electrolyte

than that measured by the TEM, indicating that the Pt NPs synthesized by the ABAR method in this work display a clean surface.

Geometric features of Pt NPs include both the size and the corresponding variation in the proportion of the surface sites. With the size variations of Pt NPs, different surface site (corner, edge, and terrace) proportions changed as well, which is another key factor related to the CO2PR. Based on the observations of the TEM images presented in Fig. 1j–m, the shapes of Pt NPs with variable sizes appear similar to each other, considering the fact that the acid or base additives cannot alter the surface energy and the growth direction of Pt NPs during the synthesis. Hence, in the absence of capping agent, the growth direction of Pt nuclei should

be isotropic to form the most stable shape in thermodynamics[32, 33]. Previous studies have shown that Pt NPs smaller than 5.0 nm tend to grow as truncated octahedron in order to minimize the surface energy[14, 34]. Therefore, to verify the structure configuration of different-sized Pt NPs, the fast Fourier transform (FFT) was adopted. For 3.4PHTSO, 4.3PHTSO, and 7.0PHTSO, Pt (111) was observed as the main exposed facet along the [100] direction (Supplementary Fig. 4 & inset and Fig. 2c & inset). Taking into account the different exposed facets, the consistent truncated octahedron model was proposed and presented. Therefore, using the same model as the reference, the proportions of different surface sites, such as Pt (111), Pt (100), corner sites, and edge sites are summarized in

Supplementary Table 2 (for calculation details, see the Supplementary Note 2)[14, 35]. Apparently, with the increase in the Pt NPs' size, the proportion of terrace sites (Pt (111) and Pt (100)) increased and the low-coordinated sites (corner and edge sites) decreased simultaneously (Fig. 2d).

**Chemical composition and electronic properties of Pt NPs.** The surface chemical states of Pt NPs were characterized by high-resolution Pt 4f XPS (X-ray photoelectron spectroscopy) (Fig. 3a–d). The calibrations made on Pt 4f XPS spectra were based on the standard binding energy of C (sp$^2$) at 284.6 eV. Generally, with the increase in binding energy, three sets of peaks located at Pt $4f_{7/2}$ and Pt $4f_{5/2}$ can be observed, which are attributed to metallic Pt$^0$ and partially oxidized Pt$^{\delta+}$ (Pt$^{2+}$ and Pt$^{4+}$) species, respectively[36]. After carefully fitting of the obtained spectra, the peaks information (full width at half maximum (FWHM), peak area, peak center, peak type, and area fraction) of Pt 4f was summarized in Supplementary Table 3. The ratio (area fraction calculated from peak integration) of Pt$^0$ was calculated to be around 67% of the sum of the Pt$^0$ and Pt$^{\delta+}$ peaks for Pt NPs of all sizes (Table 1), suggesting similar oxidation states of the Pt NPs, regardless of their size. Interestingly, with the decrease in Pt NPs' size, the binding energy of the zero-state Pt $4f_{7/2}$ exhibits a redshift to a higher region, and the 1.8-nm Pt NPs show 0.4 eV higher binding energy than that of 7.0-nm Pt NPs. In addition, the high-resolution O 1s and Ti 2p XPS spectra of $x$PHTSO ($x$ = 1.8, 3.4, 4.3, and 7.0) show the binding energies of lattice O and Ti possess almost the same values with the size variations of Pt NPs (Supplementary Fig. 5). Combined with the CO-pulse adsorption result (Table 1), we can eliminate the strong metal–support interactions among the $x$PHTSO (details in Supplementary Note 1). Previous studies have revealed that the binding energy of Pt varies in accordance with the adsorption of the adsorbate on the Pt NPs, which is determined by the numbers of low-coordinated sites in the geometric effect[37]. Consequently, this result shows that the adsorption energy of CO on the surface of Pt should follow the order of low-coordination sites greater than terrace sites, which means that based on the thermodynamics, the generated CO should preferentially occupy the low-coordination sites during the CO2PR.

The electron trapping ability is another important electronic property of Pt NPs in photocatalysis. When Pt and TiO$_2$ contacted electrically, the conduction band of TiO$_2$ bending upward and accompanied with the migration of electrons to the Pt sites until their Fermi levels are aligned. In this case, the generation of Schottky barrier at the Pt–TiO$_2$ interface would promote the photo-generated electrons' accumulation at the Pt sites and prevent the recombination of electron–hole pairs[2, 38]. In our case, the charge separation efficiency over variable-sized $x$PHTSO was characterized by the room temperature photo-luminescence (PL), and the charge transfer efficiency was assessed on the femtosecond transient absorption (fs-TA) and transient photocurrent response. The PL emission spectra are shown in Supplementary Fig. 6. At the excitation wavelength of 315 nm, the emission spectrum of HTSO shows two main strong peaks located at 370 and 470 nm that represent the charge recombination of the bandgap transition and band edge free excitation, respectively[39]. Interestingly, the weak emission spectra of $x$PHTSO ($x$ = 1.8, 3.4, 4.3, and 7.0) suggest that the charge recombination rate is greatly suppressed due to the presence of the Pt NPs. In addition, all $x$PHTSO catalysts show very similar PL intensities, indicating that the charge separations over different catalysts are similar to each other. The size-dependent charge transfer efficiency of Pt NPs was characterized by the fs-TA technique, a robust tool to track the real-time photo-

generated carrier dynamics of the nanomaterial composites[40, 41]. In the fs-TA measurement, a 310 nm pump laser beam was chosen to excite different samples according to the bandgap excitation of $x$PHTSO (Fig. 2b). Subsequently, the time-dependent photoinduced transient absorption at 350 nm and its relaxations of 1.8PHTSO and 4.3PHTSO have been fitted to two exponential decays: $A_1 \exp(-t/\tau_1) + A_2\exp(-t/\tau_2)$. The results in Fig. 3e show two sets of time constants, for 1.8PHTSO is $\tau_1 = 3.5$ ps (54.4%), $\tau_2 = 48.2$ ps (45.6%), and for 4.3PHTSO is $\tau_1 = 3.5$ ps (63.8%), $\tau_2 = 517$ ps (36.2%), respectively. According to previous studies, the faster time decay was ascribed to the trapping of the surface state and the slower time decay was originated from the recombination of the charge carriers[42–44]. By comparison, decreasing the size value of Pt NPs could effectively accelerate the slower decay of the surface charge recombination, indicates the enhanced electrons' trapping ability and extra electron transfer routes of the smaller Pt NPs. In addition, the transient photocurrent response reflects 1.8PHTSO shows an obviously higher photocurrent response than the other catalysts, due to the abundant metal–semiconductor interfaces and the lower Fermi level of smaller Pt NPs, which is beneficial for the electron transfer from HTSO to the Pt (Fig. 3f). With the increasing size of Pt NPs, the corresponding photocurrent signals decrease gradually, implying the reduced charge transfer efficiency. Hence, we can conclude that the size of Pt NPs has a distinct effect on the charge transfer efficiency but a weak influence on the charge separation efficiency.

**CO2PR evaluation result.** The dependences of the morphology, geometric features, and electronic properties on the size of Pt NPs have been carefully demonstrated, as shown above. The size variation of Pt NPs contributes to the surface site proportion, which also alters the electronic properties of the charge transfer and binding energy. Therefore, the size effect of Pt NPs in CO2PR should be studied for both the charge transfer route and the surface reaction pathway.

The evaluation of CO2PR over HTSO and $x$PHTSO ($x$ = 1.8, 3.4, 4.3, and 7.0) was performed under simulated solar light irradiation. The corresponding generation rates of each product, the reacted electrons' rate and the selectivity toward CH$_4$ are summarized in Table 2. It should be noted that the reacted electrons' rate represents the rate of photo-generation of electrons involved in the overall reaction, reflecting the charge transfer efficiency of the photocatalyst[13]. Without any modification, blank HTSO shows a very low reacted electrons' rate (2.02 μmol g$^{-1}$ h$^{-1}$), as well as a very poor performance for both activity and selectivity. As expected, after the introduction of Pt NPs as the cocatalyst, a series of $x$PHTSO show a significantly enhanced reacted electrons' rate. Decreasing the size of the Pt NPs could

**Table 2 CO2PR evaluation over HTSO and $x$PHTSO ($x$ = 1.8, 3.4, 4.3 and 7.0)$^a$**

| Sample | Production rate (μmol g$^{-1}$ h$^{-1}$)$^b$ | | | Reacted electrons' rate (μmol g$^{-1}$ h$^{-1}$)$^c$ | Selectivity for CH$_4$ (%)$^d$ |
|---|---|---|---|---|---|
| | H$_2$ | CH$_4$ | CO | | |
| HTSO | 0.39 | 0.045 | 0.44 | 2.02 | 17.8 |
| 1.8PHTSO | 58.7 | 9.7 | 1.8 | 198.6 | 39.1 |
| 3.4PHTSO | 29.7 | 7.1 | 0.80 | 117.8 | 48.2 |
| 4.3PHTSO | 16.2 | 7.2 | 0.40 | 90.8 | 63.4 |
| 7.0PHTSO | 1.1 | 1.1 | 0.062 | 11.1 | 79.1 |

$^a$ Reaction conditions were described in the method: CO2PR evaluation measurement
$^b$ Irradiation: 4 h
$^c$ The reacted electrons' rate = 8*r(CH$_4$) + 2*r(H$_2$) + 2*r(CO)
$^d$ The selectivity toward CH$_4$ = [8*r(CH$_4$)]/[2*r(CO) + 8*r(CH$_4$) + 2*r(H$_2$)]*100%

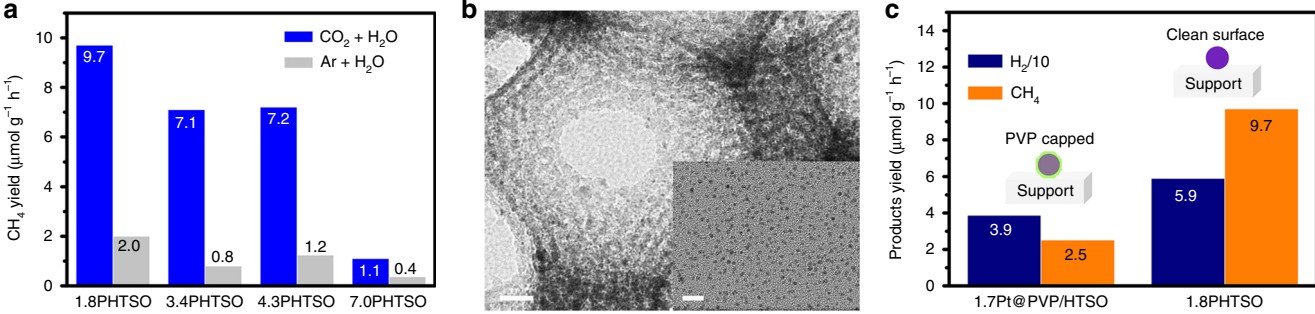

**Fig. 4** Investigation of the effect of carbon impurities in the CO2PR. **a** $CH_4$ yield comparison of $x$PHTSO ($x$ = 1.8, 3.4, 4.3, and 7.0) in different atmospheres, where blue bars denote the yield of $CH_4$ generated from $CO_2/H_2O$ atmosphere and gray bars denote the yield of $CH_4$ generated from $Ar/H_2O$ atmosphere. **b** TEM images of 1.7Pt@PVP/HTSO and 1.7 nm Pt NPs colloidal (inset) the scale bars are 20 and 10 nm (inset), and **c** comparison of $CO_2$ photoreduction evaluation over 1.7Pt@PVP/HTSO and 1.8PHTSO

drastically increase the reacted electrons' rate from 11.1 to 198.6 $\mu mol\ g^{-1}\ h^{-1}$. Meanwhile, the production rate of $CH_4$ and $H_2$ improved as well with the decrease in the Pt particle size. The 1.8PHTSO shows the highest $CH_4$ and $H_2$ yields of 9.7 and 58.7 $\mu mol\ g^{-1}\ h^{-1}$, which are 8.8 and 53.4 times higher than the yields of 7.0PHTSO, respectively. Conversely, for improving the selectivity for $CH_4$, the decreasing of the size of Pt NPs is detrimental. The most active 1.8PHTSO only shows a selectivity of 39.1% toward $CH_4$, which means that more photo-generated electrons participate in the competing HER reaction. By contrast, the selectivity of $CH_4$ is increased with the increase in the Pt NPs' size, and the highest $CH_4$ selectivity of 79.1% is achieved by the 7.0PHTSO. However, the $CH_4$ yield over 7.0PHTSO is very poor (1.1 $\mu mol\ g^{-1}\ h^{-1}$). Therefore, to obtain a higher yield and better selectivity of $CH_4$, the optimal size of the Pt should be fixed at 4.3 nm (4.3PHTSO), whereby the $CH_4$ yield can reach 7.2 $\mu mol\ g^{-1}\ h^{-1}$ and the selectivity is 63.4%.

On the other hand, the carbon impurities induced by the use of organics during the preparation are an important interference factor for the studying of CO2PR, which would modify the size effect of Pt NPs on the CO2PR performance. In our case, to verify that the products such as $CH_4$ and CO originated from $CO_2$ and $H_2O$ vapor, the control experiment was also performed under the $Ar/H_2O$ atmosphere. It was found that the yield of $CH_4$ over $x$PHTSO is negligible, as shown in Fig. 4a, indicating that $CH_4$ is predominately generated from the $CO_2$ and $H_2O$. The above CO pulse chemisorption experiment result also confirms that the as-prepared $x$PHTSO has a clean surface. In addition, to further verify the negative effect of the surface carbon impurities on the CO2PR, the 1.7Pt@PVP/HTSO was synthesized and evaluated for comparison. The obtained PVP-based composite shows a similar microstructure to that of 1.8PHTSO (Fig. 4b). However, the 1.7Pt@PVP/HTSO only shows $CH_4$ and $H_2$ yields of 2.5 and 39.0 $\mu mol\ h^{-1}$, which are 3.9 and 1.5 times lower than that of the ligand-free 1.8PHTSO, respectively (Fig. 4c). It is clear that even though the size of the Pt NPs is similar, the residue PVP covering the surface of the Pt NPs is not only blocking the surface active sites but also hindering the charge transfer route through the metal–support interface; both of these effects are detrimental for CO2PR performance.

## Discussion

The CO2PR evaluation results described above indicate that the size effect of Pt NPs plays an important role in both the activity and the selectivity. As seen from the data presented in Table 2, the variation trend of the $CH_4$ and $H_2$ yields is consistent with the trend of the reacted electrons' rate, indicating that the reaction rate of CO2PR is determined by the charge transfer efficiency.

Specifically, the size-dependent activity suggests that smaller Pt NPs favor both HER and CO2PR by enhancing the redox performance. Previous studies have shown that smaller Pt NPs (not smaller than 1.0 nm) with suitable energy levels could trap the photo-generated electrons more efficiently to enhance the charge transfer[12, 13]. Meanwhile, the higher surface density of smaller Pt NPs accompanied by more metal–support interfaces is also beneficial for the charge transfer, as confirmed by the previous fs-TA and photocurrent measurement shown in Fig. 3e, f. Therefore, more surface active sites could participate in the CO2PR and yield more products.

The size-dependent selectivity suggests that larger Pt NPs are more active in CO2PR than HER. Since $CH_4$ production is an eight-electron reaction (Eqs. 4 and 5), a sufficient supply of electrons should promote the reaction kinetics of $CH_4$ generation. Therefore, for larger Pt NPs, the paucity of electrons should not be the reason for the high $CH_4$ selectivity. In this case, the surface site proportion in the geometric feature should be considered as the main parameter that determines the selectivity for $CH_4$. As mentioned above, as the size of the Pt NPs increased, the proportion of the terrace sites increased, while that of the corner and edge sites decreased (Fig. 2d). Therefore, the correlation between the selectivity for $CH_4$ and the surface site proportion as a function of the size of Pt NPs is plotted in Fig. 5a. Clearly, the selectivity of $CH_4$ shows a consistent trend with the terrace site fraction but shows the opposite relationship with the low-coordinated site fraction over the variously sized Pt NPs. Hence, we infer that the terrace sites of the Pt NPs are the active sites for $CH_4$ generation; on the other hand, the low-coordinated sites are more active in HER.

$$CO_2 + 8e^- + 8H^+ \rightarrow CH_4 + 2H_2O. \tag{4}$$

$$H_2O + 2h^+ \rightarrow 2H^+ + 1/2O_2. \tag{5}$$

$$CO_2 + {^*} + 2e^- + 2H^+ \rightarrow {^*}CO + H_2O$$
$$(\text{asterisk denotes the surface of the catalyst}). \tag{6}$$

$$CO + {^*} \rightarrow {^*}CO. \tag{7}$$

To investigate the reactivity of different Pt surface sites in CO2PR and competing HER in thermodynamic, the density functional theory (DFT) calculations were performed (Supplementary Tables 4–6). Two classic Pt surface sites models: Pt(111) and Pt55 were built on behalf of the terrace sites and low-coordinated sites. In Fig. 5b and Supplementary Fig. 7, stepwise calculated Gibbs free energy diagrams for $CO_2$ reduction into $CH_4$ and HER were presented. Apparently, from the reaction

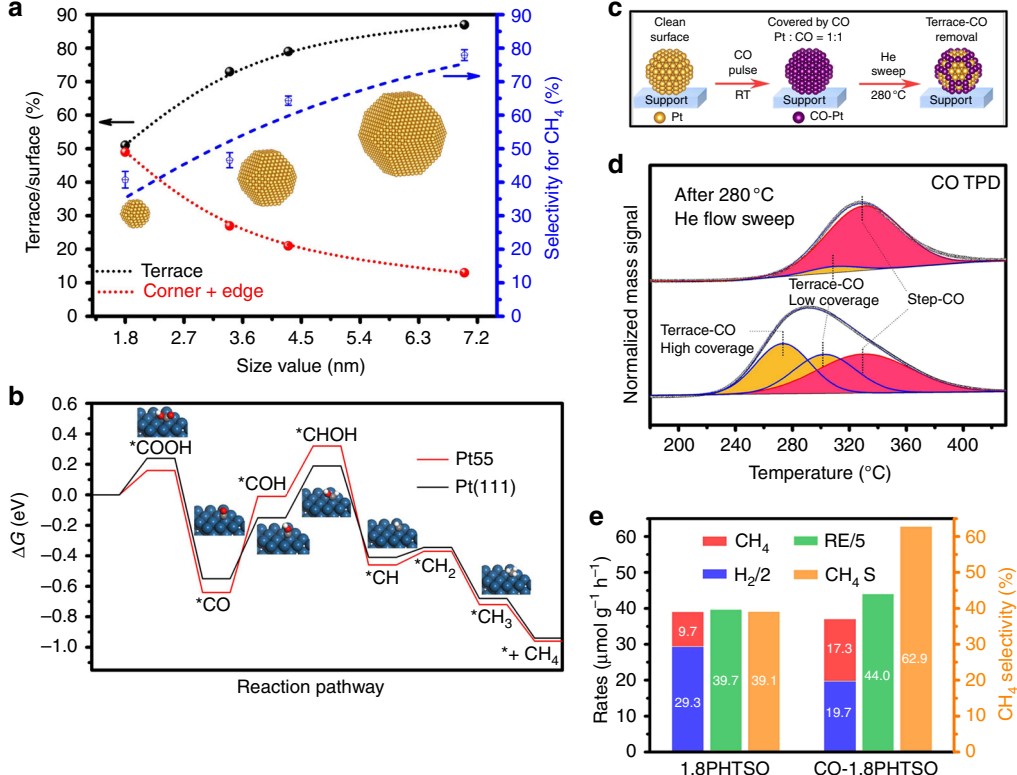

**Fig. 5** Size-dependent activity and selectivity of Pt NPs in CO2PR. **a** Correlations between the selectivity for CH$_4$ and surface site proportion as functions of the size of Pt NPs in $x$PHTSO ($x = 1.8$, 3.4, 4.3, and 7.0). **b** Free energy diagrams for CO$_2$ reduction to CH$_4$ by the thermochemical model on Pt(111) surface and Pt55. **c** Scheme illustration of partial CO-modified 1.8PHTSO through stepwise adsorption and desorption of CO. **d** CO-TPD results of 1.8PHTSO after CO pulse adsorption and stepwise CO pulse adsorption and He flow desorption at 280 °C. **e** Performance comparisons of CO2PR between 1.8PHTSO and CO-1.8PHTSO, the RE denotes as the reacted electrons, and CH$_4$ S denotes as the CH$_4$ selectivity

pathway we noticed that the following hydrogenation of *CO and *COH in the third and fourth steps with the highest energy barrier would be the rate limiting steps. In this case, Pt(111) with obvious lower energy barrier (0.74 eV) compared with Pt55 (0.96 eV) in the rate limiting steps, which means Pt(111) possessed with higher catalytic activity in the CO2PR toward CH$_4$ (Fig. 5b). Moreover, in the competing HER reaction, Pt55 outperformed Pt (111) with lower energy barrier, suggests its higher activity toward HER (Supplementary Fig. 7).

The separated roles of low-coordinated sites and terrace sites in HER and CO2PR are further discussed based on their distinctly different adsorption energies toward CO. It should be noted that regardless of the Pt NPs' sizes, the production rate of CO is much lower than that of CH$_4$ (Table 2). This is despite the fact that CO generation is a two electron reaction (Eq. 6), which should be more facile to proceed compared with the generation of CH$_4$[5, 21]. However, due to the strong interaction between the CO molecules and Pt atoms (Supplementary Fig. 8), we believe that most of the CO molecules are hardly desorbed from Pt, or are generated from TiO$_2$ and are recaptured by Pt (Eqs. 6 and 7), so that even the bare HTSO yields more CO than 4.3PHTSO and 7.0PHTSO (Table 2). In this case, with the photocatalytic reaction proceeding, the strongly bonded CO will deactivate the Pt active sites gradually if it cannot be protonated to yield CH$_4$ in time, and that would explain why many Pt NP-based photocatalysts show relative high activity but low stability in CO2PR[12, 22, 45]. Supplementary Fig. 9 shows the time evolution of three major products over $x$PHTSO ($x = 1.8$, 3.4, 4.3, and 7.0) in 4 h. It is found that the stability of HER is reduced when the size of the Pt NPs decreases. For 1.8PHTSO, the production rate of H$_2$ declines from 83.9 μmol g$^{-1}$ h$^{-1}$ at the first hour down to 36.3 μmol g$^{-1}$ h

$^{-1}$ at the fourth hour; meanwhile, the CH$_4$ yield is less affected. Therefore, considering that the binding energy of low-coordinated sites is stronger than the terrace sites, the generated CO should preferentially adsorb on the low-coordinated sites based on thermodynamics (Supplementary Fig. 8); on the contrary, the terrace sites of Pt NPs are less occupied for the low CO* coverage, and hence, the H$_2$ and CH$_4$ yields show distinct differences over time. The cyclic experiments of 1.8PHTSO further prove this hypothesis (Supplementary Fig. 10). The results show that the average H$_2$ production rate in 4 h declines from 58.7 μmol g$^{-1}$ h$^{-1}$ at the first run to 46.1 μmol g$^{-1}$ h$^{-1}$ at the third run. On the contrary, the CH$_4$ yield is slightly improved after the first run, and consequently, the selectivity for CH$_4$ improved from 39.1% (first run) to 48.1% (third run). Although the catalyst was reactivated at 180 °C before each cyclic test, the strongly bonded CO may not be removed completely and may gradually accumulate. To further verify the above speculation, we proposed a series CO temperature programmed desorption experiment (CO-TPD) on 1.8PHTSO to selective keeping low-coordinated sites bonded CO and removing terrace sites bonded CO (Fig. 5c, see Methods section for more information). From the CO-TPD curve, three desorption peaks were observed, which assigned to terrace-CO at high coverage, the terrace-CO at low coverage and the step-CO, respectively (Fig. 5d)[15]. Therefore, we choose 280 °C as the critical point to selective remove the terrace-CO in He flow (Fig. 5d), thereafter, the sample with partial CO-modified 1.8PHTSO (denotes as CO-1.8PHTSO) was collected and used immediately for the CO2PR evaluation. In CO2PR, the CO-1.8PHTSO shows obviously higher CH$_4$ selectivity up to 62.9% compared with the 39.1% of 1.8PHTSO, which means the competitive HER in CO-1.8PHTSO has been suppressed effectively

(Fig. 5e). The production yield of $CH_4$ over CO-1.8PHTSO also increases from 9.7 to 17.3 μmol g$^{-1}$ h$^{-1}$. Besides, the unchanged particle size and dispersity of Pt NPs in CO-1.8PHTSO indicates the treatment of 280 °C He flow sweep cannot alter the original structure of 1.8PHTSO (Supplementary Fig. 11). Meanwhile, the similar reacted electrons' rate of these two samples indicates the CO modification did not affect the efficiency of Pt NPs in charge separation dynamics, but greatly boosting the CO2PR for the selective $CH_4$ formation (Fig. 5e). Combined with the DFT calculation results, we can firmly conclude the separate roles of terrace sites and low-coordinated sites represent as the active sites for the CO2PR and competitive HER reaction respectively. Beyond that, we also offered a promising surface CO modification strategy on smaller Pt NPs to acquire higher activity and selectivity toward $CH_4$ simultaneously.

Without any decoration, the optimum size of Pt NPs in our case should be the 4.3 nm, which the 4.3PHTSO presents both higher activity and selectivity for $CH_4$. Since $H_2O$ is the sole electron donor in CO2PR, the competing HER reaction is inevitable. Therefore, the possible solution for simultaneously acquiring high activity and selectivity toward $CH_4$ should be the rational surface modification over Pt NPs to suppress the low-coordinated sites from participating in the HER reaction, or to improve the chemical adsorption amount toward $CO_2$ over $H_2O$ at the terrace active sites.

In summary, the ligand-free Pt NPs with a size ranging from 1.8 to 7.0 nm dispersed well on the HTSO have been successfully realized. The size manipulation of Pt NPs was achieved by adjustment of the nucleation amounts through the addition of acidic or basic additives, and the ABAR synthesis process can be easily spread to other materials. In CO2PR, the Pt NPs show a prominent size effect for both activity and selectivity. Smaller Pt NPs with greater metal–semiconductor interface area promote the charge transfer efficiency and thus exhibit superior activity in $CH_4$ and $H_2$ generation. The higher $CH_4$ selectivity was achieved by the larger Pt NPs due to the higher fraction of terrace sites. The terrace sites of Pt NPs were found to be the active sites for $CH_4$ generation, and the low-coordinated sites are more favorable in the HER and could be gradually deactivated by CO. Our work may provide insight into the development of highly efficient Pt NP-based cocatalysts in semiconductor-based CO2PR.

## Methods

**General information.** The chemicals we used in this research, including styrene ($C_8H_8$, AR), sodium dodecyl sulfate ($C_{12}H_{25}SO_4Na$, SDS, AR), potassium persulfate ($K_2S_2O_8$, KPS, AR), sodium hydroxide (NaOH, AR), tetraethyl orthosilicate ($C_8H_{20}O_4Si$, TEOS, AR), titanium tetrachloride ($TiCl_4$, AR), titanium isopropoxide ($C_{12}H_{28}O_4Ti$, TTIP, AR), pluronic triblock copolymer ($EO_{20}PO_{70}EO_{20}$, abbreviated as P123, $M_{av} = 5800$, Aldrich), ethanol (EtOH, AR), MeOH (AR), chloroplatinic acid hexahydrate ($H_2PtCl_6·6H_2O$, Pt > 37.5%), EG (AR), HCl (AR), sodium hydroxide (NaOH, AR), polyvinyl pyrrolidone (K-30, PVP, AR), Evonik P25 (commercial), carbon nitride ($C_3N_4$) nanosheet and ultrapure water were purchased without any further purification.

**Preparation of catalysts.** The hierarchical ordered $TiO_2$–$SiO_2$ porous material (HTSO) was synthesized following the classic evaporation induced self-assemble method with two templates[8]. Generally, the precursor of Ti–Si sol was synthesized by the mixing of P123 (1.0 g), EtOH (15 mL), TEOS (0.45 mL), $TiCl_4$ (0.26 mL), and TTIP (1.66 mL). The Ti–Si sol was stirring at room temperature for 2 h, and then, the pre-dried poly-styrene bulk were impregnated with the Ti–Si sol in a petri dish. The petri dish was transferred into an oven (the relative humidity is set at 50–60%) to sequentially age at 40 and 70 °C for 3 days. After aging treatment, the impregnated PS bulk were calcinated in a muffle furnace at 500 °C for 4 h, and the obtained product is collected and denoted as HTSO. The obtained HTSO was used as the supportive semiconductor for Pt growth without any modifications. Four sized Pt NPs in situ growth on the HTSO was realized through an ABAR method. During the synthesis process, EG or MeOH acting as both the solvent and reducing agent, $H_2PtCl_6·6H_2O$ was used as the Pt precursor, the Pt loading amount was kept at 2 wt% for all samples, NaOH–EG solution and different amount of HCl–EG solution were used to create basic or acidic environment. With the increase of the

Pt NPs size, different samples were denoted as $x$PHTSO ($x$ represented the Pt particle size: 1.8, 3.4, 4.3, and 7.0 nm) based on the TEM result. If not mentioned, the $x$PHTSO was used as the joint name of the four samples. Typically, for 1.8PHTSO synthesis, 50 mL EG and 0.2 g HTSO were mixed in a three-neck flask and sonicated for 15 min, then 2.1 mL $H_2PtCl_6·6H_2O$ EG solution (5 mg mL$^{-1}$) was added to the flask. After 30 min stirring, the flask was evacuated and filled with $N_2$ and 1.25 mL–0.25 M NaOH–EG solution was injected subsequently, the flask was heated at 433 K for 2 h, after cooling down to room temperature, 2 mL–0.25 M HCl–EG solution was added and followed another 2 h stirring. For 3.4PHTSO synthesis, 50 mL methanol and 0.2 g HTSO were mixed and sonicated for 15 min, 2.1 mL $H_2PtCl_6·6H_2O$ EG solution (5 mg mL$^{-1}$) was added. After 30 min stirring, the mixture was refluxed for 3 h. For 4.3PHTSO and 7.0PHTSO synthesis, similar process was followed according to 1.8PHTSO synthesis, only 1 and 4 mL 0.25 M HCl–EG solution were added respectively instead of NaOH–EG solution. Similarly, the different-sized Pt NPs supported on Evonik P25 or $C_3N_4$ sheet also followed with this procedure; just simply replace the HTSO to P25 or $C_3N_4$ sheet. The 1.7Pt@PVP/HTSO sample was synthesized by the impregnation method. Firstly, 1.7 nm Pt NPs colloidal was synthesized in advanced: 4 mL $H_2PtCl_6·6H_2O$–EG solution in a three-neck flask with $N_2$ inlet, 1.25 mL–0.5 M NaOH–EG solution was injected into the flask and solution was heated in 433 K for 2 h, after cooling down to room temperature, 2 mL–0.5 M HCl solution was added and the mixture was centrifuged to remove the supernatant and redispersed in PVP–ethanol solution (5 mL ethanol and 0.025 g PVP K-30). Then, 0.1 g HTSO was dispersed in 20 mL ethanol with sonication; 0.84 mL Pt colloidal was added dropwise into the mixture and followed with another 12 h stirring. After the alcohol reduction reaction, and the impregnation, all of the samples were washed thoroughly with ethanol and $H_2O$ respectively, finally, the samples were dried in vacuum oven overnight at 60 °C.

**Photocatalysts characterizations.** The structures of samples were analyzed by a TEM (JEM-2100) equipped with FFT; the crystallinity of samples were characterized by XRD (Rigaku D/MAX 2550, Cu K radiation, $\lambda = 1.5406$ Å), whose operation voltage and current was set at 40 kV and 40 mA. The porous structures of samples (mean pore size distribution and the BET surface area) were characterized by an ASAP2020 instrument at 77 K. The surface chemical states of Pt in different samples were characterized by high-resolution XPS (Perkin-Elmer PHI 5000C ESCA system: Al $K_\alpha$ radiation, operated at 250 W), and the shift of the spectra caused by the relative surface charging was calibrated according to the standard binding energy of C (sp$^2$) at 284.6 eV. ICP-AES analysis was adopted on the Agilent 725ES to characterize the real loading amount of Pt in samples, prior to each test, the $x$PHTSO ($x$ = 1.8, 3.4, 4.3, and 7.0) powder was etching in the nitrohydrochloric acid/hydrofluoric acid mixture. The charge separation efficiency of samples was evaluated on a luminescence spectrometry (Cary Eclipse). In PL emission spectra, the excitation wavelength is set at 315 nm. The charge transfer efficiency of samples was evaluated by transient photocurrent response. The tests were conducted on a Zennium electrochemical station with a three electrode cell. The working electrode is made of FTO glass and as-prepared sample, the counter electrode and the reference electrode is a Pt wire and a saturated calomel electrode, respectively. The $Na_2SO_4$ aqueous solution (0.5 M) is used as electrolyte and a 300 W Xeon lamp is used as the light source. The CO pulse chemisorption was carried out on the Micrometritics AutoChem II 2920 chemisorption analyzer with helium (He) as the carrier gas and mass spectrum as the detector. Prior to each test, 50 mg as-prepared $x$PHTSO ($x$ = 1.8, 3.4, 4.3, and 7.0) sample was reduced in the $H_2$/He (1:9) flow at 473 K for 30 min, then the gas flow was switched into pure He at 473 K and last for 30 min, afterwards, the sample was cooled down to room temperature in the He flow. The CO pulse was performed in the use of 1% CO and He balance gaseous mixture, every 2 min, 0.1573 mL CO/He mixture was pulsed into the sample cell, the test was ceased until CO was no longer consumed from the obtained the pulse curves. The stoichiometry between the adsorbed CO molecules and Pt atoms was assumed as 1:1; therefore, the total surface exposed Pt atoms ($Pt_{surface}$) were equaled with the CO adsorption amount. In addition, the total atoms of Pt ($Pt_{total}$) in each sample was calculated based on Eq. 8, $\omega_{Pt}$ denotes as the weight percentage of Pt from the ICP-AES result and $N_A$ denotes as the Avogadro constant. Therefore, the metal dispersion ($D$) of Pt in each sample was calculated based on the Eq. 9. Lastly, the particle size is calculated by using Eq. 10, with the premise of spherical Pt NPs assumption and the Pt atom density at $1.25 \times 10^{19}$ atoms m$^{-2}$[27].

$$Pt_{total} = N_A^* 1^* \omega_{Pt}/M_{Pt}. \tag{8}$$

$$D(\%) = Pt_{surface}/Pt_{total}^* 100\%. \tag{9}$$

$$d(nm) = 1.13/D. \tag{10}$$

The fs-TA measurements were performed based on a femtosecond Ti:Sapphire regenerative amplified Ti:Sapphire laser system (Coherent, Astrella-Tunable-F-1k) and fs-TA spectrometer system (Ultrafast Systems, Helios Fire). The fs-TA experimental setup and methods was provided here[46]. Fs-TA measurements were

carried out by using a femtosecond regenerative amplified Ti:Sapphire laser system. The amplifier has been seeded with the 844 fs laser pulses, which comes from an oscillator laser system. The amplified 800 nm laser pulses (~15%) would produce a white-light continuum (320–700 nm) in a CaF$_2$ crystal to produce the probe laser pulse. This probe beam would continuously split into two beams before going through the specimen. One probe laser beam traverses the specimen. Meanwhile, another probe laser beam goes to the reference spectrometer to monitor the fluctuations of the probe beam intensity. For experiments described in this work, the 1.8PHTSO and 4.3PHTSO emulsion were dispersed on the quartz plates to form thin films, the films should be transparent to make sure the probe beam can pass through the quartz plate. The transparent films were then excited by a 310 nm pump laser beam (4 mW) which was produced by a femtosecond mode TOPAS. The TOPAS was pumped by ~30% of the amplified 800 nm laser pulses.

**CO2PR evaluation measurement**. The gas–solid phase CO2PR experiment was conducted in a home-made evaluation system. First, 30 mg as synthesized sample was dispersed in water homogeneously under sonication, then the above emulsion was dried in oven to form a thin layer deposited on the petri dish. After that, the petri dish was transferred into the reactor (300 mL volume with a quartz cover) and placed on a glass support. Under the glass support, there is a magnetic fan-shaped stir and its purpose is to circulate the gases in reactor homogeneously. Subsequently, after 30 min vacuum treatment of the reactor, 1 bar CO$_2$ (gas, 99.995%) and 1 mL H$_2$O were introduced into the reactor successively. Under the light irradiation (a 300 W Xeon lamp coupled with an AM 1.5 filter was used as the light source), the gaseous mixture were analyzed by a gas chromatograph (GC, Shimadzu GC-2014) every 1 h, carbon involved products, such as methanol, methane, and carbon monoxide, were detected by the flame ionization detector, other species, such as hydrogen, oxygen, nitrogen, carbon dioxide, etc., were detected by the thermal conductivity detector. The qualitative and quantitative analyses of the gaseous products were based on the external method taking the concentrated standard gases as reference. The data reproducibility were checked by performing the same reaction in duplicate. To verify the origination of carbon involved species, three sets of control tests were conducted: (1) performing the evaluation test in the Ar and H$_2$O atmosphere instead of CO$_2$ and H$_2$O atmosphere; (2) performing the evaluation test in the absence of photocatalyst; and (3) performing the evaluation test in the absence of light irradiation.

**Theoretical calculations**. DFT calculations were performed using the Vienna Ab Initio Simulation package within gradient (generalized gradient approximation-Perdew-Burke-Ernzerhof (GGA-PBE)). A plane-wave basis set with a cutoff energy of 350 eV and ultrasoft Vanderbilt pseudopotentials was employed. The Brillouin zone integration was carried out with $4 \times 4 \times 1$ k-point sampling for Pt(111), and single k-point (gamma point) was used for Pt55. The Pt(111) surface was chosen to represent the terrace sites, while Pt55 cluster was chosen to represent the corner sites of Pt nanoparticles. For Pd model surfaces, four layers $(3 \times 3)$ Pt(111) was used, while the slab is separated from its periodic images in the vertical direction by a vacuum space of 20 Å. During the geometry optimizations, only two bottom layers were fixed while the upper two layers were fully relaxed. The Pt55 cluster was cut with cuboctahedral and truncated-octahedron shapes in orthorhombic boxes. All atoms and adsorbates of Pt clusters were fully relaxed during geometry optimization. In free energies calculations, the integrated heat capacity, entropy corrections, and zero-point energy (ZPE) have been included[47–49].

Take reaction 3 as an example, to calculate the free energy change from adsorbed CO to adsorbed COH, the free energy change of the below chemical reaction needs to be calculated:

$$^*CO + (H^+ + e^-) \rightarrow {}^*COH. \quad (11)$$

The free energy change of this reaction would be:

$$\Delta G_{S3} = G(^*COH) - G(^*CO) - \mu(H^+) - \mu(e^-)$$
$$= G(^*COH) - G(^*CO) - \tfrac{1}{2}\mu(H_{2(g)}) + eU, \quad (12)$$

where $U$ is the potential relative to the electrode at the same conditions as the reactions. The chemical potentials of each adsorbed species were calculated by standard DFT techniques below. The free energy for one species could be calculated by the standard formula:

$$G = E_{DFT} + ZPE + \delta H_0 - TS, \quad (13)$$

where $E_{DFT}$, ZPE, $\delta H_0$, and $TS$ are the total energy from DFT calculations, the zero-point energy, the integrated heat capacity, and the product of the temperature $(T)$, and the entropy. Equation 14 can be rewritten as

$$\Delta G_{S3} = \Delta E(^*COH) - \Delta E(^*CO) + \Delta ZPE + \Delta\delta H_0 - T\Delta S + eU. \quad (14)$$

The relevant thermodynamic data of gas-phase species are given in Supplementary Table 4. The relevant data of gas-phase species, adsorbates, substrates, and free energies ($\Delta G$) under the standard conditions and 0 V are given

in Supplementary Tables 5 and 6. Free energy diagrams are shown in Fig. 5b and Supplementary Fig. 7.

**Data availability**. The authors declare that the data supporting the findings of this study are available within the article and the Supplementary Information files.

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

## Acknowledgements

This work was supported by the National Natural Science Foundation of China (21773062, 21577036, 21377038, 5171101651), the State Key Research Development Program of China (2016YFA0204200), the Shanghai Education Development Foundation and Shanghai Municipal Education Commission (16JC1401400), the Shanghai Pujiang Program (17PJD011), and the Fundamental Research Funds for the Central Universities (22A201514021).

## Author contributions

C.D. and C.L. contributed equally to this work. C.D. and M.X. conceived and designed the research. C.D. synthesized photocatalysts and conducted all the experiments. C.L. and H.L. carried out the theoretical calculations. M.L. performed the fs-TA test. C.D. and M.X. wrote the paper. S.H., Z.D., J.G., and J.Z. gave suggestions on the experiment and writing. All authors discussed and analyzed the data.

## Additional information

**Competing interests:** The authors declare no competing interests.

