## [Peer Review File(PDF 2242 kb) · Nature Communications]

Reviewers' comments:

Reviewer #1 (Remarks to the Author):

This is a rather good paper studying size effects on CO₂ photoreduction on platinum nanoparticles. The experiments seem competently performed. One of the problems in this kind of studies stems from the difficulty in separating true size effect from other interferences such as small amounts of uncontrolled contamination on the surface. However, in the present case the authors seem to have been very careful to demonstrate that the observed trends are real. The results are conveniently discussed and the conclusions are very relevant for the scientific community. Therefore, I recommend the publication of this ms. as it is. There are only a few mistypes that need to be corrected before final publication. For instance, when describing light absorption experiments, "adsorption" should be replaced by "absorption". Also at the end of the section describing the preparation of the catalyst, there are a couple of "was" that should be "were".

Reviewer #2 (Remarks to the Author):

This study deals with the effect of Pt nanoparticles size on the activity and selectivity of CO₂ photocatalytic reduction. In this study various samples composed of platinum nanoparticles deposited on TiO₂-SiO₂ porous materials were synthesized. For each sample, mass metal loadings as well as the size of Pt nanoparticles are well controlled. The synthesis of particles is well described. Morphological, structural and electronic properties of platinum nanoparticles are studied. Finally, the size effect was studied both on charge transfer efficiency and reaction pathway. The manuscript is well-written and experimental results are well described and commented. They will have a broad impact for the community. Nevertheless some points need to be clarified. For all the above mentioned reason I recommend a minor revision for this manuscript.

1 The authors state that there is a close contact between Pt Nps and TiO₂-based substrate. The effect of the interface between Pt nanoparticles and oxide substrate on the electronic properties of platinum nanoparticles has not been discussed and results concerning electronic properties of platinum nanoparticles are only explained by size effect. Is there any SMSI effect?

2 All the discussion is based on the assumption that the shape of platinum nanoparticles is truncated octahedron. On HRTEM images provided in Fig 1. (j-m) it is not obvious that Pt nanoparticles have the same shape whatever the considered sample. Is there any defects such as twins or stacking faults ?

3 XPS spectra should be better discussed. It seems that there is an evolution in the respective amounts of Pt²⁺ and Pt⁴⁺ species from sample to sample. This should be explained. Moreover it seems that for 1.8PHTSO sample, photopeak ascribed to metallic Pt⁰ has a higher full width at half maximum than for other samples. The authors should give an explanation for that.

Reviewer #3 (Remarks to the Author):

This paper reports the size-dependent activity and selectivity of CO₂ photocatalytic reduction over Pt nanoparticles. In fact, similar works about the size-dependent activity and selectivity in the CO₂ reduction reaction have been reported in various metals. Compared with the previous reports, the present works did not show significantly important advance for CO₂ reduction. In addition, there is no

strong supports for the charge transfer efficiency, the terrace sites as the active sites, and the low-coordinated sites for H₂ evolution reaction selectivity of CO₂PR.

Response to Referee 1

We appreciate the reviewer for their generous recommendation of our manuscript for publication in *Nature Communications*.

Comment 1.

There are only a few mistypes that need to be corrected before final publication. For instance, when describing light absorption experiments, “adsorption” should be replaced by “absorption”. Also at the end of the section describing the preparation of the catalyst, there are a couple of “was” that should be “were”.

Answer: We thank the reviewer for the kindly comment, we have revised all the mistakes you have mentioned in the Revised Manuscript. For instance, in line 151: “light adsorption feature of xPHTSO” has been corrected to “light absorption feature of xPHTSO”, and the saying in line 448: “all of the samples was washed thoroughly” has been corrected to “all of the samples were washed thoroughly”.

Revision made: *Line 68:* cause the imbalance of light absorption.....;

Line 133: single-sized Pt NPs are highly dispersed on the surface of C₃N₄ and P25.....;

Lines 154-156: characterize the light absorption..... enhanced visible light absorption..... due to the light absorption.....;

Line 158: However, the unchanged absorption.....;

Line 160: Additionally, the related flat absorption.....;

Lines 501-503: NaOH-EG solution and different amount of HCl-EG solution weredifferent samples were.....;

Line 515: only 1mL and 4mL 0.25M HCl-EG solution were.....;

Lines 527-528: all of the samples were the samples were.....;

Response to Referee 2

We would like to thank the reviewer for the careful reading of our manuscript. The suggestions and comments have been carefully considered and our work has been revised to address the points the reviewer has raised, as shown below. We hope our revision can make the paper much more acceptable for publication in *Nature Communications*.

Comment 1.

The authors state that there is a close contact between Pt Nps and TiO₂-based substrate. The effect of the interface between Pt nanoparticles and oxide substrate on the electronic properties of platinum nanoparticles has not been discussed and results concerning electronic properties of platinum nanoparticles are only explained by size effect. Is there any SMSI effect?

Answer: We thank the reviewer for the suggestion. Firstly, we would like to explain the saying of “close contact between the Pt and TiO₂”. In fact, different from the PVP-stabilized Pt NPs supported on HTSO, the ligand-free Pt NPs supported on HTSO obtained through our synthesis strategy would result in the generation of direct metal-oxide contact interface (Figs. 1j~m), which is beneficial to the photo-generated electrons’ separation and transfer. What’s more, there’s no doubt that the metal-oxide interface plays important roles in photocatalysis, prior studies have shown that when metal NPs and semiconductor contacted electrically (usually the work function of metal is higher than that of semiconductor), the conduction band of semiconductor bending upward and accompanied with the migration of electrons to the metal until their Fermi levels are aligned. In this case, the Schottky barrier could be formed at the metal-semiconductor interface to promote the photo-generated electrons’ accumulation at the metal sites and prevent the recombination of electron-hole pairs^{1, 2}. So, in our case, the Pt and HTSO (TiO₂) represent the metal and semiconductor respectively and we have also added more discussion about the electronic properties of the interface between Pt-TiO₂ in the revised manuscript.

According to previous studies, generally, the strong interactions could be generated between the group VIII metals and the reducible metal-oxide supports (such as TiO₂) in H₂ atmosphere at a high temperature^{3, 4}. Subsequently, the formed strong metal-support interactions (SMSI) would result in

two distinct features: 1) electrons transferred from the support to the metal sites and result in an increased electron density at the metal sites; 2) the adsorption towards CO and H₂ at the metal sites decreased dramatically due to the encapsulation of metal NPs by the support^{5,6}. Amiridis et al. once studied the effects of reduction temperature and SMSI on Pt/TiO₂ composites, the results indicate that the H₂ pre-treatment of Pt/TiO₂ at a low reduction temperature (200 °C or less) cannot affect its CO adsorption behavior, however, when H₂ pre-treatment of Pt/TiO₂ exceeds 300 °C, the adsorption amount towards CO could be reduced dramatically⁷. Therefore, in our case, consider of the relative consistent and gentle preparation method of the xPHTSO (x = 1.8, 3.4, 4.3 and 7.0), the reduction temperature of Pt precursor is only 160 °C in ethylene glycol (EG), hence, it can be eliminated the SMSI effect between Pt and HTSO. Besides, CO-pulse adsorption result indicates the adsorption amount of CO is highly dependent on the size value of Pt NPs, in addition, the calculated size values based on the adsorption result are correlated with the result from the TEM measurement. These results could demonstrate the Pt NPs are not encapsulated by the support and the mutual interactions are not strong like the SMSI effect. To further exclude this possibility in our case, the high-resolution O 1s and Ti 2p XPS spectra of xPHTSO (x = 1.8, 3.4, 4.3 and 7.0) were added in **Supplementary Fig. 5**. The result shows that the binding energies of lattice O and Ti possess almost the same value with the size variation of Pt NPs, which indicates the interactions between Pt NPs and TiO₂ in xPHTSO (x = 1.8, 3.4, 4.3 and 7.0) are equivalent, increase or decrease the size of Pt NPs couldn't change the mutual interactions between Pt and TiO₂^{8,9}. On the contrary, with the decrease of the size of Pt NPs, the obvious red-shift of Pt 4f binding energy to a higher region further indicates the binding energy alterations of Pt 4f in the xPHTSO result from the geometric properties of Pt NPs instead of the different metal-support interactions (**Fig. 3a-d**). Hence, we can eliminate the SMSI effect from our case, the size effect of Pt NPs is the main reason for the activity and selectivity variation in CO₂PR. Based on the above explanation, we made corresponding revisions in the revised manuscript.

Supplementary Figure 5 | High-resolution O 1s and Ti 2p XPS spectra of (a, e) 1.8PHTSO, (b, f) 3.4PHTSO, (c, g) 4.3PHTSO and (d, h) 7.0PHTSO.

Revision made: *Lines 265-269:* When Pt and TiO₂ contacted electrically, the conduction band of TiO₂ bending upward and accompanied with the migration of electrons to the Pt sites until their Fermi levels are aligned. In this case, the Schottky barrier could be formed at the Pt-TiO₂ interface to promote the photo-generated electrons' accumulation at the Pt sites and prevent the recombination of electron-hole pairs.

Lines 243-248: In addition, the high-resolution O 1s and Ti 2p XPS spectra of xPHTSO (x = 1.8, 3.4, 4.3 and 7.0) shows the binding energies of lattice O and Ti possess almost the same values with the size variations of Pt NPs (Supplementary Fig. 5). Combined with the CO-pulse adsorption result (Table 1), we can eliminate the strong metal-support interactions among the xPHTSO (details in Supplementary Information).

Comment 2.

All the discussion is based on the assumption that the shape of platinum nanoparticles is truncated octahedron. On HRTEM images provided in Fig 1. (j-m) it is not obvious that Pt nanoparticles have the same shape whatever the considered sample. Is there any defects such as twins or stacking faults?

Answer: We thank the suggestion from the reviewer. Firstly, in order to verify the consistent truncated octahedron configuration existed in other sizes of Pt NPs, the measurement of HR-TEM coupled with FFT was performed on the 3.4PHTSO and 4.3PHTSO and the results were added in the **Supplementary Fig. 4**. Clearly, both Pt NPs in 3.4PHTSO and 4.3PHTSO show truncated octahedron shape and the Pt (111) was observed along the [100] direction, which is well agreed with the result of 7.0PHTSO in Fig. 2c. Actually, in order to study the surface sites variation of different sized Pt NPs, the truncated octahedron was selected as the prototype according to previous researches^{10, 11}. When the particle size of Pt is less than 5nm, the surface of Pt was mainly covered by (111) and (100) facets in order to minimize the surface energy¹⁰. Besides, in the previous studies, Xia and co-workers once declared that the twinned nanostructures are rarely formed in Pt nanocrystals probably due to the high internal strain energies associated with these structures and the occurrence of oxidative etching (due to O₂) during a typical synthesis^{12, 13}. Due to the twin defects and stacking faults are common features in the oriented growth mechanism^{14, 15, 16}, their existence could promote the anisotropic growth of fcc metals¹³. However, in the acid-base mediate alcohol reduction (ABAR) process, none of the capping agent was involved. In addition, in the process of TEM measurement and particle sizes statistics, we found that most of the Pt NPs show spherical-like shape, other shapes such as nanowires, nanorods or nanoplates haven't been observed in our case. Thereby, we believe the growth direction of Pt nuclei should be isotropic and form the most stable shape in thermodynamics: truncated octahedrons.

Supplementary Figure 4 | (a, d) HR-TEM images of 3.4PHTSO and 4.3PHTSO, where the red

squares represent the local magnification of one Pt NP (b, e), the marked facets show the classic truncated octahedron shape of Pt NP. (c, f) The corresponding FFT patterns of Pt NP. The scale bars are 10nm in (a, d) and 2nm in (b, e).

Revision made: *Line 215-217:* Hence, in the absence of the capping agent, the growth direction of Pt nuclei should be isotropic to form the most stable shape in thermodynamics.

Line 220-222: For 3.4PHTSO, 4.3PHTSO and 7.0PHTSO, Pt (111) was observed as the main exposed facet along the [100] direction (Supplementary Fig. 4 & inset and Fig. 2c & inset).

Comment 3.

XPS spectra should be better discussed. It seems that there is an evolution in the respective amounts of Pt²⁺ and Pt⁴⁺ species from sample to sample. This should be explained. Moreover, it seems that for 1.8PHTSO sample, photopeak ascribed to metallic Pt⁰ has a higher full width at half maximum than for other samples. The authors should give an explanation for that.

Answer: We thank the reviewer for the significant comment. Firstly, to exclude the deviations that may come from the XPS measurement as well as improving the accuracy of the results, we repeated all the XPS tests of xPHTSO (x = 1.8, 3.4, 4.3 and 7.0) samples at one time. After the very careful tests, all the XPS spectra were carefully fitted. The revised Pt 4f XPS spectra are shown in Figs. 3a~d and the peak information summary (FWHM, peak area, peak center, peak type, and area fraction) is also shown in **Supplementary Table 3**. It should be noted that the revised Pt 4f XPS spectra didn't alter the red-shift phenomenon for 1.8PHTSO, the binding energies of zero state Pt 4f_{7/2} of xPHTSO remain the same. What's more, the FWHM values of Pt⁰, Pt²⁺, Pt⁴⁺ show consistent results (± 0.05) for all xPHTSO (x = 1.8, 3.4, 4.3 and 7.0) samples. The revised area fraction of Pt⁰ in xPHTSO (x = 1.8, 3.4, 4.3 and 7.0) is 69.8%, 67.1%, 67.3% and 67.1% respectively (the Table 2 in the manuscript also has been updated), which indicates the similar oxidation states of Pt in xPHTSO (x = 1.8, 3.4, 4.3 and 7.0).

Revised Figure 3 | Electronic properties of xPHTSO. High-resolution Pt 4f XPS spectra of (a) 1.8PHTSO, (b) 3.4PHTSO, (c) 4.3PHTSO and (d) 7.0PHTSO. (e) The kinetics of the characteristic transient absorption band observed at 350 nm in the fs-TA spectra after 310 nm excitation of samples 1.8PHTSO and 4.3PHTSO. The solid lines were the curves fitted by a two-exponential. (f) Transient photocurrent response of xPHTSO (x = 1.8, 3.4, 4.3 and 7.0), where a 300-W Xe lamp is used as the light source and a 0.5 M Na₂SO₄ solution is used as the electrolyte.

Supplementary Table 3 | Peak information summary from high-resolution Pt 4f XPS spectra of xPHTSO (x = 1.8, 3.4, 4.3 and 7.0).

Peak Info Sample	FWHM	Peak Area	Peak Center (eV)	Peak Type	Area Fraction (%)
1.8PHTSO	1.08	810	70.7 (Pt 4f 7/2 Pt ⁰)	Gaussian	Pt ⁰ (69.8) Pt ²⁺ (16.8) Pt ⁴⁺ (13.3)
	0.94	150	71.6 (Pt 4f 7/2 Pt ²⁺)		
	1.40	100	72.4 (Pt 4f 7/2 Pt ⁴⁺)		
	1.08	660	74.0 (Pt 4f 5/2 Pt ⁰)		
	0.95	205	75.0 (Pt 4f 5/2 Pt ²⁺)		
	1.4	180	76.0 (Pt 4f 5/2 Pt ⁴⁺)		
3.4PHTSO	0.98	1004	70.4 (Pt 4f 7/2 Pt ⁰)	Gaussian	Pt ⁰ (67.3) Pt ²⁺ (14.0) Pt ⁴⁺ (18.7)
	0.90	260	71.3 (Pt 4f 7/2 Pt ²⁺)		
	1.5	143	72.4 (Pt 4f 7/2 Pt ⁴⁺)		
	1.03	940	73.7 (Pt 4f 5/2 Pt ⁰)		
	0.90	300	74.7 (Pt 4f 5/2 Pt ²⁺)		
	1.5	241	75.8 (Pt 4f 5/2 Pt ⁴⁺)		
4.3PHTSO	1.01	760	70.4 (Pt 4f 7/2 Pt ⁰)	Gaussian	Pt ⁰ (67.1) Pt ²⁺ (19.3) Pt ⁴⁺ (13.6)
	0.90	220	71.3 (Pt 4f 7/2 Pt ²⁺)		
	1.40	103	72.3 (Pt 4f 7/2 Pt ⁴⁺)		
	1.05	726	73.7 (Pt 4f 5/2 Pt ⁰)		
	0.90	227	74.7 (Pt 4f 5/2 Pt ²⁺)		
	1.40	180	75.8 (Pt 4f 5/2 Pt ⁴⁺)		
7.0PHTSO	0.99	270	70.3 (Pt 4f 7/2 Pt ⁰)	Gaussian	Pt ⁰ (67.1) Pt ²⁺ (18.7) Pt ⁴⁺ (14.2)
	0.96	60	71.1 (Pt 4f 7/2 Pt ²⁺)		
	1.5	35	71.8 (Pt 4f 7/2 Pt ⁴⁺)		
	0.98	250	73.6 (Pt 4f 5/2 Pt ⁰)		
	0.96	85	74.6 (Pt 4f 5/2 Pt ²⁺)		
	1.5	75	75.6 (Pt 4f 5/2 Pt ⁴⁺)		

Revision made: *Line 236-241:* After carefully fitting of the obtained spectra, the peaks information (FWHM, peak area, peak center, peak type, and area fraction) of Pt 4f was summarized in Supplementary Table 3. The ratio (area fraction calculated from peak integration) of Pt⁰ was calculated to be around 67% of the sum of the Pt⁰ and Pt^{δ+} peaks for Pt NPs of all sizes (Table 1), suggesting similar oxidation states of the Pt NPs regardless of their size.

Response to Reviewer 3

We would like to thank the reviewer for the careful reading of our manuscript. We are so sorry that we cannot substantially display the scientific significance of our manuscript to the reviewer due to our limitations of expression. We deeply respect the reviewer's comments and advice, so, we have added the follow-up research work in the Revised Manuscript to highlight the significantly important advance for CO₂ photoreduction of our research work. We hope our very careful revision can make our paper acceptable for publication in *Nature Communications*.

Comment 1.

In fact, similar works about the size-dependent activity and selectivity in the CO₂ reduction reaction have been reported in various metals. Compared with the previous reports, the present works did not show significantly important advance for CO₂ reduction.

Answer: We thank the reviewer for asking these questions.

Firstly, we would like to further demonstrate the novelty of our research works. In the gas-phase CO₂ photocatalytic reduction (CO₂PR) reaction, the electrons with specific reduction potential were generated from the semiconductor then transferred to the Pt sites and trigger the reduction reaction, which is very different from the working mechanism of liquid-phase CO₂RR reaction. Besides, the reduction pathways of metal NPs based CO₂RR mainly focused on the CO and H₂ evolution^{17, 18, 19, 20}, the study of surface active sites of Pt NPs in CO₂ reduction toward CH₄ driven by the photocatalysis and their size dependent activity and selectivity between CH₄ and H₂ has never been reported. In our research, in order to objectively study the size effect of Pt NPs in CO₂PR, we for the first time proposed an acid-base mediate alcohol reduction (ABAR) strategy to synthesize size controllable Pt NPs in the absence of organic stabilizing agent and at consistent loading amount of Pt precursor. The size dependent alterations of geometric features and electronic properties in Pt NPs have been illustrated in detail. In CO₂PR, for the first time we demonstrate the intrinsic size-dependent activity and selectivity of Pt NPs, the terrace sites and low-coordinated sites acting as the active sites for CH₄ and H₂ evolution respectively. **Our results have showed that it is difficult to synchronously increase the yield and selectivity of CH₄ in CO₂PR by regulating Pt size. Hence, we added the follow-up research work in the Revised Manuscript to achieve the**

synchronous increase of the yield and selectivity of CH₄ in CO₂PR over the small sized Pt cocatalytic system, which would further confirm the significantly important advance for CO₂ photoreduction compared with the previous reports.

Supplementary Figure 8 | Adsorption energy of CO on (a) Pt(111), (b) Pt55.

Previously studies have shown that the CO molecules could be strongly adsorbed on the surface of Pt NPs, the desorption temperatures are correlated with the binding strength of CO on different surface sites of Pt^{21, 22}. In our case, the DFT calculation result indicates that the Pt55 has a higher adsorption energy of CO than Pt(111) (-1.72 eV vs -1.52 eV), as shown in **Supplementary Figure 8**. So, in the CO temperature programmed desorption (CO-TPD) experiment on Pt NPs, the pre-adsorbed CO molecules would desorb from the surface of Pt at different temperature ranges separately, that is, the terrace-CO would desorb before the step-CO. So, inspired by this phenomenon, we propose that if we could just remove the terrace-CO in 1.8PHTSO, the rest step-CO would act as the blocking agents to prevent the low-coordinated sites from participating in the CO₂PR or HER reaction (illustrated in **Fig. 5c**), in this case, we can study the performance of 1.8PHTSO in CO₂PR with only terrace site exposed.

Revised Figure 5 | (a) Correlations between the selectivity for CH₄ and surface site proportion as functions of the size of Pt NPs in xPHTSO ($x = 1.8, 3.4, 4.3$ and 7.0). (b) Free energy diagrams for CO₂ reduction to CH₄ by the thermochemical model on Pt(111) surface and Pt55. (c) Scheme illustration of partial CO modified 1.8PHTSO through stepwise adsorption and desorption of CO. (d) CO-TPD results of 1.8PHTSO after CO pulse adsorption (up) and stepwise CO pulse adsorption and He flow desorption at 280 °C (down). (e) Performance comparisons of CO₂PR between 1.8PHTSO and CO-1.8PHTSO, the RE denotes as the reacted electrons and CH₄ S denotes as the CH₄ selectivity.

To obtain the desorption temperatures of terrace-CO and the step CO-modified 1.8PHTSO (abbreviates as CO-1.8PHTSO) photocatalyst, three sets of independent CO-adsorption and TPD experiments were conducted on 1.8PHTSO. Firstly, the CO-adsorption TPD test was performed and the result is shown in revised **Fig. 5d** (down). It shows three distinct desorption peaks, which represent the terrace-CO at high coverage, the terrace-CO at low coverage and the step-CO respectively. So, we choose 280 °C as the critical point to remove most of the terrace-CO solely. Subsequently, in the second test, the CO-TPD was performed after the pre-treatment of CO adsorbed

1.8PHTSO in helium (He) flow sweep at 280 °C, the result shows that most of the terrace-CO has been removed but the step-CO was left. Thereafter, in the third test, the CO was pre-adsorbed on the 1.8PHTSO, once the terrace-CO was removed at 280 °C in the He flow, the partial CO modified 1.8PHTSO (denotes as CO-1.8PHTSO) was collected and used immediately for the CO₂PR evaluation (**Fig. 5c**) (the detailed description of the CO-TPD was shown in the revised **Supplementary Information**). To exclude the possibility of the particle sinter and size increase of CO-1.8PHTSO, considered of the 280 °C He flow sweep treatment, the HR-TEM analysis was conducted on the CO-1.8PHTSO and the result was shown in **Supplementary Fig. 11**. The unchanged particle size and dispersity of Pt NPs in CO-1.8PHTSO indicates the treatment of 280 °C He flow sweep couldn't alter the original structure of 1.8PHTSO.

Supplementary Figure 11 | TEM and corresponding HR-TEM images of 1.8PHTSO after CO pulse adsorption and 280 °C He flow sweep, the red squares indicate the stepwise magnification of local sites, the red arrows point out the dispersive Pt NPs. The scale bar is 100nm in (a), 10nm in (b) and 5nm in (c).

As a result, in CO₂PR, the CO-1.8PHTSO shows obviously higher CH₄ selectivity up to 62.9% compared with the 39.1% of 1.8PHTSO, which means the competitive HER in CO-1.8PHTSO has been suppressed effectively (**Fig. 5e**). Meanwhile, the production yield of CH₄ over CO-1.8PHTSO also increases from 9.7 to 17.3 μmol/g·h. Notably, the similar reacted electrons' rate of these two samples indicates the CO modification couldn't alter the efficiency of Pt NPs in charge separation dynamics, but greatly boosted the CO₂PR for the selective CH₄ generation. Therefore, we can conclude the separate roles of terrace sites and low-coordinated sites represent the active sites for the CO₂PR and competitive HER reaction respectively. Besides, we offered a promising partial CO modification strategy on smaller Pt NPs to acquire higher activity and selectivity towards CH₄ in

CO₂PR simultaneously.

Based on the above results, we strongly believe our research presents novelty in both materials synthesis and reaction mechanism explanations. And it also has significantly important advance for CO₂ reduction compared with the previous reports. We believe our work provides novel insights into the development of highly efficient Pt involved photocatalysts in CO₂PR.

In sum, we conclude the novelty of our research as follows.

(1) First use of ABAR method to achieve the controlling of Pt NPs from 1.8 to 7.0nm in the absence of polyvinylpyrrolidone (PVP) and other carbon impurities, which first time enables the precise control of the size of Pt NPs under the assumption of a constant loading amount, and exclusion of the negative influence of the carbon impurities induced by the use of PVP during the preparation.

(2) First demonstration of an objective and in-depth understanding of the Pt NPs' size effect in the CO₂PR. Decreasing the size of Pt NPs promotes the charge transfer efficiency and thus enhances the CO₂PR and HER activity, but leads to higher selectivity towards H₂ over CH₄. The higher CH₄ selectivity is achieved by the larger Pt NPs due to the higher fraction of terrace sites.

(3) First certification of the active sites in Pt NPs cocatalytic CO₂PR. The terrace sites of Pt NPs are found to be the active sites for CH₄ generation, and the low-coordinated sites are more favorable in the HER and could be gradually deactivated by CO.

(4) First making it clear that the selective passivation of low-coordinated sites over small sized Pt NPs is indeed a very useful way to increase the yield and selectivity of methane at one time.

Comment 2:

In addition, there is no strong supports for the charge transfer efficiency, the terrace sites as the active sites, and the low-coordinated sites for H₂ evolution reaction selectivity of CO₂PR.

Answer: Firstly, in order to characterize the size-dependent charge transfer efficiency of Pt NPs, we adopt femtosecond transient absorption (fs-TA) technique, a robust tool to track the real-time photoexcited carrier dynamics of the nanomaterial composites^{23, 24}. In the fs-TA measurement, a 310nm pump laser beam was chosen to excite different samples according to the bandgap excitation of xPHTSO (**Fig. 2b**). Subsequently, the time-dependent photoinduced transient absorption at 350nm

and its relaxations of 1.8PHTSO and 4.3PHTSO have been fitted to two-exponential decays: $A_1 \exp(-t/\tau_1) + A_2 \exp(-t/\tau_2)$. The result in **Fig. 3e** shows two sets of time constants, for 1.8PHTSO is $\tau_1 = 3.5\text{ps}$ (54.4%), $\tau_2 = 48.2\text{ps}$ (45.6%) and for 4.3PHTSO is $\tau_1 = 3.5\text{ps}$ (63.8%), $\tau_2 = 517\text{ps}$ (36.2%) respectively. According to previous studies, the faster time decay was ascribed to the trapping of the surface state and the slower time decay was originated from the recombination of the charge carriers^{25, 26, 27}. By comparison, decreasing the size value of Pt NPs could effectively accelerate the slower decay of the surface charge recombination, indicates the enhanced electrons' trapping ability and extra electron transfer routes of the smaller Pt NPs.

Revised Figure 3 | Electronic properties of xPHTSO. High-resolution Pt 4f XPS spectra of (a) 1.8PHTSO, (b) 3.4PHTSO, (c) 4.3PHTSO and (d) 7.0PHTSO. (e) The kinetics of the characteristic transient absorption band observed at 350 nm in the fs-TA spectra after 310 nm excitation of samples 1.8PHTSO and 4.3PHTSO. The solid lines were the curves fitted by a two-exponential. (f) Transient photocurrent response of xPHTSO (x = 1.8, 3.4, 4.3 and 7.0), where a 300-W Xe lamp is used as the

light source and a 0.5 M Na₂SO₄ solution is used as the electrolyte.

Supplemental Table 4 | Thermodynamic data of gas-phase species. Zero-point energies (ZPE) are calculated with experimental vibrational data, the integrated heat capacity (δH_0) and entropy (S) at 298.15K are obtained from reference. For water, the entropy is calculated at 0.035bar, because at this pressure gas-phase H₂O is in equilibrium with liquid water at 298.15K.

Adsorbate	E_{elec} (ev)	ZPE(ev)	δH_0 (ev)	-TS(ev)	μ (ev)
H ₂	-0.00	0.27	0.09	-0.41	-0.05
CO	1.75	0.13	0.09	-0.62	1.35
CO ₂	0.9	0.31	0.10	-0.66	0.65
H ₂ O	0.03	0.56	0.10	-0.68	0.01
CH ₄	-1.22	1.20	0.10	-0.65	-0.56

Supplemental Table 5 | Contributions to the adsorbate free energy on Pt(111) from the zero-point energy correction, enthalpic temperature correction, entropy and the total free energy correction respectively. ΔE is the electronic energy of the state minus the electronic energy of the clean slab(s) associated with that state.

Adsorbate	ΔE (ev)	ZPE(ev)	δH_0 (ev)	-TS(ev)	ΔG (ev)
*COOH 	0.43	0.60	0.01	-0.18	0.24
*CO 	0.23	0.18	0.01	-0.15	-0.55
*COH 	1.17	0.44	0.01	-0.18	-0.15

*CHOH		1.16	0.76	0.01	-0.19	0.19
*CH		1.10	0.47	0.01	-0.17	-0.41
*CH ₂		0.67	0.49	0.01	-0.17	-0.34
*CH ₃		-0.08	0.50	0.01	-0.17	-0.68

Supplemental Table 6 | Contributions to the adsorbate free energy on Pt55 from the zero-point energy correction, enthalpic temperature correction, entropy and the total free energy correction respectively. ΔE is the electronic energy of the state minus the electronic energy of the clean slab(s) associated with that state.

Adsorbate		ΔE (ev)	ZPE(ev)	δH_0 (ev)	-TS(ev)	ΔG (ev)
*COOH		0.35	0.60	0.01	-0.18	0.16
*CO		0.03	0.18	0.01	-0.15	-0.64
*COH		1.24	0.44	0.01	-0.18	-0.01
*CHOH		1.28	0.76	0.01	-0.19	0.32
*CH		1.02	0.47	0.01	-0.17	-0.46
*CH ₂		0.61	0.49	0.01	-0.17	-0.37

-0.16

0.50

0.01

-0.17

-0.72

Supplementary Figure 7 | Calculated free energy diagram for H^+ reduction to H_2 by the thermochemical model on Pt(111) surface and Pt55.

Secondly, to investigate the reactivity of different Pt surface sites in CO₂PR and competing HER in thermodynamics, the density functional theory (DFT) calculations were performed (details in **Supplemental Tables 4~6**). Two classic Pt surface sites models: Pt(111) and Pt55 were built on behalf of the terrace sites and low-coordinated sites. In **Fig. 5b** and **Supplementary Fig. 7**, the stepwise calculated Gibbs free energy diagrams for CO₂ reduction into CH₄ and HER were presented. Apparently, from the reaction pathway of CO₂PR we noticed that the following hydrogenation of CO* and COH* in the third and fourth steps with the highest energy barrier would be the rate limiting steps. In this case, the Pt(111) with obvious lower energy barrier (0.74 eV) compared with Pt55 (0.96 eV) in the rate limiting steps, which means the Pt(111) possessed with higher catalytic activity in the CO₂PR towards CH₄ (**Fig. 5b**). Moreover, in the competing HER reaction, Pt55 outperformed Pt(111) with lower energy barrier, suggests its higher activity towards HER (**Supplementary Fig. 7**).

To sum up, we adopt the fs-TA measurement to characterize the size-dependent charge transfer efficiency of Pt NPs, the result is in accordance with the result from transient photocurrent response, which both suggests the smaller Pt NPs are beneficial for the charge transfer. Meanwhile, in

thermodynamics, the DFT calculations clearly demonstrate the Pt(111) (represents the terrace sites) is the active site towards CH₄, on the other hand, Pt55 (represents the low-coordinated sites) is more active in HER.

Revision made: *Lines 271-272:* the charge transfer efficiency was assessed on the femtosecond transient absorption (fs-TA) and transient photocurrent response.

Lines 280-296: The size-dependent charge transfer efficiency of Pt NPs was characterized by the femtosecond transient absorption (fs-TA) technique, a robust tool to track the real-time photoexcited carrier dynamics of the nanomaterial composites^{23, 24}. In the fs-TA measurement, a 310nm pump laser beam was chosen to excite different samples according to the bandgap excitation of xPHTSO (Fig. 2b). Subsequently, the time-dependent photoinduced transient absorption at 350nm and its relaxations of 1.8PHTSO and 4.3PHTSO have been fitted to two-exponential decays: $A_1 \exp(-t/\tau_1) + A_2 \exp(-t/\tau_2)$. The results in Fig. 3e shows two sets of time constants, for 1.8PHTSO is $\tau_1 = 3.5\text{ps}$ (54.4%), $\tau_2 = 48.2\text{ps}$ (45.6%) and for 4.3PHTSO is $\tau_1 = 3.5\text{ps}$ (63.8%), $\tau_2 = 517\text{ps}$ (36.2%) respectively. According to previous studies, the faster time decay was ascribed to the trapping of the surface state and the slower time decay was originated from the recombination of the charge carriers^{25, 26, 27}. By comparison, decreasing the size value of Pt NPs could effectively accelerate the slower decay of the surface charge recombination, indicates the enhanced electrons' trapping ability and extra electron transfer routes of the smaller Pt NPs. In addition, the transient photocurrent response reflects 1.8PHTSO shows an obviously higher photocurrent response than the other catalysts...

Lines 406-418: To investigate the reactivity of different Pt surface sites in CO₂PR and competing HER in thermodynamic, the density functional theory (DFT) calculations were performed (details in Supplemental Tables 4~6). Two classic Pt surface sites models: Pt(111) and Pt55 were built on behalf of the terrace sites and low-coordinated sites. In Fig. 5b and Supplementary Fig. 7, stepwise calculated Gibbs free energy diagrams for CO₂ reduction into CH₄ and HER were presented. Apparently, from the reaction pathway we noticed that the following hydrogenation of CO* and COH* in the third and fourth steps with the highest energy barrier would be the rate limiting steps. In

this case, Pt(111) with obvious lower energy barrier (0.74 eV) compared with Pt55 (0.96 eV) in the rate limiting steps, which means Pt(111) possessed with higher catalytic activity in the CO₂PR towards CH₄ (Fig. 5b). Moreover, in the competing HER reaction, Pt55 outperformed Pt(111) with lower energy barrier, suggests its higher activity towards HER (Supplementary Fig. 7).

Lines 448-470: To further verify the above speculation, we proposed a series CO temperature programmed desorption experiment (CO-TPD) on 1.8PHTSO to selective keeping low-coordinated sites bonded CO and removing terrace sites bonded CO (Fig. 5c, see Supplementary Information for details). From the CO-TPD curve, three desorption peaks were observed, which assigned to terrace-CO at high coverage, the terrace-CO at low coverage and the step-CO respectively (Fig. 5d)²¹. Therefore, we choose 280 °C as the critical point to selective remove the terrace-CO in He flow (Fig. 5d), thereafter, the sample with partial CO modified 1.8PHTSO (denotes as CO-1.8PHTSO) was collected and used immediately for the CO₂PR evaluation. In CO₂PR, the CO-1.8PHTSO shows obviously higher CH₄ selectivity up to 62.9% compared with the 39.1% of 1.8PHTSO, which means the competitive HER in CO-1.8PHTSO has been suppressed effectively (Fig. 5e). Besides, the unchanged particle size and dispersity of Pt NPs in CO-1.8PHTSO indicates the treatment of 280 °C He flow sweep cannot alter the original structure of 1.8PHTSO (Supplementary Fig. 11). Meanwhile, the similar reacted electrons' rate of these two samples indicates the CO modification didn't affect the efficiency of Pt NPs in charge separation dynamics, but greatly boosting the CO₂PR for the selective CH₄ formation (Fig. 5e). Combined with the DFT calculation results, we can firmly conclude the separate roles of terrace sites and low-coordinated sites represent as the active sites for the CO₂PR and competitive HER reaction respectively. Beyond that, we also offered a promising surface CO modification strategy on smaller Pt NPs to acquire higher activity and selectivity towards CH₄ simultaneously.

References

1. Amy L. Linsebigler, Guangquan Lu, John T. Yates J. Photocatalysis on TiO₂ surfaces: Principles, mechanisms, and selected results. *Chem. Rev.* **95**, 735-758 (1995).
2. Habisreutinger SN, Schmidt-Mende L, Stolarczyk JK. Photocatalytic reduction of CO₂ on TiO₂ and other semiconductors. *Angew. Chem., Int. Ed.* **52**, 7372-7408 (2013).
3. S. J. Tauster, S. C. Fung, Garten RL. Strong metal-supportinteractions. Group 8 noble metals

- supported on TiO₂. *J. Am. Chem. Soc.* **100**, 170 (1978).
4. S. J. Tauster, S. C. Fung, R. T. K. Baker, Horsley JA. Strong interactions in supported-metal catalysts. *Science* **211**, 1121 (1981).
 5. Matsubu JC, *et al.* Adsorbate-mediated strong metal-support interactions in oxide-supported Rh catalysts. *Nat. Chem.* **9**, 120-127 (2017).
 6. Liu X, *et al.* Strong metal-support interactions between gold nanoparticles and ZnO nanorods in CO oxidation. *J. Am. Chem. Soc.* **134**, 1121-1125 (2012).
 7. Oleg S. Alexeev, Soo Yin Chin, Mark H. Engelhard, Lorna Ortiz-Soto, Amiridis MD. Effects of reduction temperature and metal-support interactions on the catalytic activity of Pt/ γ -Al₂O₃ and Pt/TiO₂ for the oxidation of CO in the presence and absence of H₂. *J. Phys. Chem. B* **109**, 23430-23443 (2005).
 8. Wangcheng Zhan, *et al.* A sacrificial coating strategy toward enhancement of metal–support interaction for ultrastable Au nanocatalysts. *J. Am. Chem. Soc.* **138**, 16130-16139 (2016).
 9. Xing Cheng, *et al.* Highly active, stable oxidized platinum clusters as electrocatalysts for the hydrogen evolution reaction. *Energy Environ. Sci.* **10**, 2450-2458 (2017).
 10. Byungkwon Lim, *et al.* Pd-Pt bimetallic nanodendrites with high activity for oxygen reduction. *Science* **324**, 1302-1305 (2009).
 11. Hyunjoon Song, Franklin Kim, Stephen Connor, Gabor A. Somorjai, Yang P. Pt nanocrystals: Shape control and langmuir-blodgett monolayer formation. *J. Phys. Chem. B* **109**, 188-193 (2005).
 12. Chen J, Lim B, Lee EP, Xia Y. Shape-controlled synthesis of platinum nanocrystals for catalytic and electrocatalytic applications. *Nano Today* **4**, 81-95 (2009).
 13. Xia Y, Xiong Y, Lim B, Skrabalak SE. Shape-controlled synthesis of metal nanocrystals: simple chemistry meets complex physics? *Angew. Chem., Int. Ed.* **48**, 60-103 (2009).
 14. Lim B, Kobayashi H, Camargo PHC, Allard LF, Liu J, Xia Y. New insights into the growth mechanism and surface structure of palladium nanocrystals. *Nano Res.* **3**, 180-188 (2010).
 15. Sean Maksimuk, Xiaowei Teng, Yang H. Roles of twin defects in the formation of platinum multipod nanocrystals. *J. Phys. Chem. C* **111**, 14312-14319 (2007).
 16. Jung Ho Yu, *et al.* Synthesis of quantum-sized cubic ZnS nanorods by the oriented attachment mechanism. *J. Am. Chem. Soc.* **127**, 5662-5670 (2005).
 17. Gao D, *et al.* Size-dependent electrocatalytic reduction of CO₂ over Pd nanoparticles. *J. Am. Chem. Soc.* **137**, 4288-4291 (2015).
 18. Mistry H, *et al.* Exceptional size-dependent activity enhancement in the electroreduction of CO₂ over Au nanoparticles. *J. Am. Chem. Soc.* **136**, 16473-16476 (2014).
 19. Back S, Yeom MS, Jung Y. Active sites of Au and Ag nanoparticle catalysts for CO₂ electroreduction to CO. *ACS Catal.* **5**, 5089-5096 (2015).
 20. Liu S, *et al.* Shape-dependent electrocatalytic reduction of CO₂ to CO on triangular silver nanoplates. *J. Am. Chem. Soc.* **139**, 2160-2163 (2017).
 21. Perez-Alonso FJ, *et al.* The effect of size on the oxygen electroreduction activity of mass-selected platinum nanoparticles. *Angew. Chem., Int. Ed.* **51**, 4641-4643 (2012).
 22. J. M, Lundwall, Sean M. McClure, Goodman DW. Probing terrace and step sites on Pt nanoparticles using CO and ethylene. *J. Phys. Chem. C* **114**, 7904-7912 (2010).
 23. Xiao J-D, *et al.* Boosting photocatalytic hydrogen production of a metal–organic framework decorated with platinum nanoparticles: The platinum location matters. *Angew. Chem., Int. Ed.* **128**,

9535-9539.

24. Li X, *et al.* Single-atom Pt as co-catalyst for enhanced photocatalytic H₂ evolution. *Adv. Mater.* **28**, 2427-2431 (2016).
25. Kim SM, *et al.* Hot carrier-driven catalytic reactions on Pt-CdSe-Pt nanodumbbells and Pt/GaN under light irradiation. *Nano Lett.* **13**, 1352-1358 (2013).
26. Wheeler DA, Zhang JZ. Exciton dynamics in semiconductor nanocrystals. *Adv. Mater.* **25**, 2878-2896 (2013).
27. Du J, *et al.* Zn-Cu-In-Se quantum dot solar cells with a certified power conversion efficiency of 11.6%. *J. Am. Chem. Soc.* **138**, 4201-4209 (2016).

REVIEWERS' COMMENTS:

Reviewer #2 (Remarks to the Author):

All modifications have been made and I think this manuscript can now be published in Nature Communications.

Reviewer #3 (Remarks to the Author):

This paper has greatly been improved and can be accepted now.

Response to Referee 2

Comment: All modifications have been made and I think this manuscript can now be published in Nature Communications.

Answer: We appreciate the reviewer for their generous recommendation of our manuscript for publication in *Nature Communications*.

Response to Reviewer 3

Comment: This paper has greatly been improved and can be accepted now.

Answer: We thank the reviewer for the acceptance of our manuscript for publication in *Nature Communications*.